# Changes in environmental and engineered conditions alter the plasma membrane lipidome of fractured shale bacteria

Chika Jude Ugwuodo,[1,2] Fabrizio Colosimo,[3] Jishnu Adhikari,[4] Kent Bloodsworth,[5] Stephanie A. Wright,[5] Josie Eder,[5] Paula J. Mouser[2]

**ABSTRACT** Microorganisms that persist in fractured shale reservoirs cause several problems including secreting foul gases and forming biofilms. Current biocontrol measures often fail due to limited knowledge of their *in situ* activities. The plasma membrane protects the cell, mediates many of its critical functions, and responds to intracellular cues and ecological perturbations through physicochemical modifications. As such, it provides valuable insight into the physiological adaptation of microorganisms in disturbed environmental systems. Here, we (i) demonstrate how changes in salinity and hydraulic retention time (HRT) influence the plasma membrane intact polar lipid (IPL) chemistry of model bacterium, *Halanaerobium congolense* WG10, and mixed microbial consortia enriched from shale-produced fluids and (ii) elucidate adjustments in membrane IPL chemistry during biofilm growth relative to planktonic cells. We incubated *H. congolense* WG10 in chemostats under three salinities (7%, 13%, and 20% NaCl), operated under three HRTs (19.2, 24, and 48 h), and in drip flow biofilm reactors under the same salinity gradients. Also, mixed microbial consortia in produced fluids were enriched in triplicate chemostat vessels under three HRTs (19.2, 24, and 72 h) and biofilm reactors. Lipids were analyzed by ultra high performance liquid chromatography-tandem mass spectrometry (UPLC-MS/MS). Our results show that phosphatidylglycerols, cardiolipins, and phosphatidylethanolamines were predominantly enriched in planktonic *H. congolense* WG10 cells grown at hypersalinity (20%) compared to optimum (13%). In addition, several zwitterionic phosphatidylcholines and phosphatidylethanolamines were higher in abundance during biofilm growth. These observations suggest that microbial adaptation and biofilm formation in fractured shale are enabled by strategic plasma membrane IPL chemistry adjustments.

**IMPORTANCE** Microorganisms inadvertently introduced into the shale reservoir during fracturing face multiple stressors including brine-level salinities and starvation. However, some anaerobic halotolerant bacteria adapt and persist for long periods of time. They produce hydrogen sulfide, which sours the reservoir and corrodes engineering infrastructure. In addition, they form biofilms on rock matrices, which decrease shale permeability and clog fracture networks. These reduce well productivity and increase extraction costs. Under stress, microbes remodel their plasma membrane to optimize its roles in protection and mediating cellular processes such as signaling, transport, and energy metabolism. Hence, by observing changes in the membrane lipidome of model shale bacteria, *Halanaerobium congolense* WG10, and mixed consortia enriched from produced fluids under varying subsurface conditions and growth modes, we provide insight that advances our knowledge of the fractured shale biosystem. We also offer data-driven recommendations for improving biocontrol efficacy and the efficiency of energy recovery from unconventional formations.

Address correspondence to Paula J. Mouser, Paula.Mouser@unh.edu.

Fabrizio Colosimo is currently employed by New England Biolabs, Ipswich, MA, United States. Jishnu Adhikari is currently employed by Tetra Tech Inc., King of Prussia, PA, United States. The remaining authors declare no conflict of interest.

See the funding table on p. 18.

**KEYWORDS** membrane adaptation, *Halanaerobium*, fractured shale, intact polar lipids, lipidomics, salinity, hydraulic retention time

Some microorganisms that are inadvertently introduced into the deep biosphere during hydraulic fracturing to extract natural gas and oil from shale formations survive numerous stressors and persist for long periods of time (1–4). The persisting microbial communities in engineered shale reservoirs are mostly comprised of anaerobic halophilic and halotolerant taxa (5, 6). *Halanaerobium* has been found to be ubiquitous and dominate many geologically distinct formations (3, 6). The composition of the shale microbiome is, however, variable and dependent on several spatiotemporal factors including intrinsic geochemistry, engineering parameters, and production duration. These bacteria become key drivers of subsurface biogeochemistry through secretion of metabolic by-products such as the well foulant, hydrogen sulfide (7), and formation of biofilms on fracture surfaces (8). Hydrogen sulfide is produced through sulfate reduction by sulfate-reducing bacteria including *Halanaerobium* (7). This gas alters the composition of the reservoir and also facilitates corrosion of engineering infrastructure (9). On the one hand, cellular aggregation on shale matrices further reduces the permeability of the rock and clogs its microfractures, which are gas flow channels. These decrease the efficiency of hydrocarbon extraction. On the other hand, some microorganisms in fractured shale wells enhance secondary natural gas recovery through methanogenesis. This usually involves metabolic synergy between bacterial fermenters and archaeal methanogens. In one well-described instance, glycine betaine (GB) fermentation by *Halanaerobium* yields trimethylamine, which is consumed by *Methanohalophilus* to produce methane (3, 10).

A comprehensive understanding of the complex microbiology of fractured shale reservoirs is very important to the global energy sector. In the US, shale accounted for over 79% of dry natural gas production in 2020 and is projected to continue to meet most of the dry natural gas needs of the country through 2050 (11, 12). Unfortunately, knowledge of *in situ* growth kinetics, metabolisms, and physiological adaptations of shale bacteria is still limited despite recent breakthroughs (1, 3, 10), which constrains effective biocontrol and efforts to harness their beneficial potential. The plasma membrane (PM) protects the cell, mediates many of its critical functions (e.g., materials exchange, energy metabolism, and surface attachment) (13), and responds to intracellular cues and ecological perturbations through physicochemical modifications (14, 15). As such, it holds valuable insights into the behavior and physiological adaptations of microorganisms in disturbed environmental systems such as engineered shale, which is largely untapped.

Fractured shale is an extreme environment for microbial growth and, by virtue of drilling/hydraulic fracturing, a highly disturbed ecosystem. It is characterized by elevated temperatures, nutrient limitation, anoxia, elevated pressures, and brine-level salinities (1, 5, 16). For instance, the salinity of flowback and produced water, which is co-collected with natural gas, could increase more than fourfold up to 6,000 ppm in just about 30 days post-fracturing (17, 18). This is due to the mixing of produced fluids with formation of brine as well as dissolution of minerals and salts on fracture surfaces (19–21). In addition, hydraulic retention time (HRT), which is the length of time that injected fluids reside belowground before recovery at the surface, broadly varies and influences carbon flux and availability. These injected fluids inadvertently carry fresh-water bacteria that must adapt to the deep shale environment by making regular fitting morphological and physiological adjustments to cope with these dynamic stresses. This involves remodeling the membrane lipidome to maintain or acquire optimal physicochemical states such as those needed for rewiring intracellular metabolism or forming biofilms. Intact polar lipid (IPL) chemistry is a key determinant of plasma membrane biophysics and functions and is, thus, especially adjusted accordingly (14, 15). Membrane IPLs differ based on headgroup net charge; hence, there are neutral, anionic (negatively charged), and zwitterionic lipids, which have a balance of positive and negative charges.

Bacterial membranes are most commonly comprised of phosphatidylethanolamine (PE), phosphatidylglycerol (PG), and cardiolipin (CL). Other less common lipids are phosphatidylcholine (PC), phosphatidylserine, phosphatidylinositol, ornithine lipids, diacylglycerol (DG), triacylglycerol (TG), glycolipids, and sphingolipids. The IPL composition largely depends on taxa and cell envelope structure. PE comprises about 75% of all phospholipids (PLs) in *Escherichia coli* and other Gram-negative bacteria (22). Variations in IPL chemical and physical properties, due to differences in headgroup and acyl chain compositions, greatly influence the biophysical attributes and functions of the bacterial plasma membrane.

Evidence suggests that some bacteria increase the amount of negatively charged intact polar lipids in their plasma membrane in response to increasing salinities (23, 24). This is due to their bilayer-stabilizing properties and implicated role in osmoprotection (24–27). Bacterial plasma membrane lipidomic changes in response to changing conditions are, however, very diverse and still being unraveled across taxa and environments (14). To the best of our knowledge, there is no existing study on how changes in salinity and hydraulic retention time affect membrane IPL chemistry in shale reservoir bacteria, and the implications of these changes on their activities and system biogeochemistry are still elusive. In addition, there is little information on the cellular membrane lipidome adjustments underlying biofilm formation on fracture surfaces. Biofilms are a major concern for engineering subsurface hydrocarbon reservoirs—they exacerbate biocontrol failure, clog fractures, and diminish the operational performance of engineering infrastructure (7, 8, 28). Therefore, understanding how changing subsurface conditions influence biofilm biology, which is mediated by the plasma membrane, is crucial for more efficient natural resource extraction.

Here, by coupling laboratory experiments with field investigations, we unravel how changes in salinity (environmental) and hydraulic retention time (engineered) influence the plasma membrane IPL chemistry of model bacterium, *Halanaerobium congolense* WG10, and mixed microbial consortia enriched from shale-produced fluids. We also elucidate adjustments in membrane IPL chemistry during biofilm growth relative to planktonic cells. We link these nanoscale events to ecosystem-scale reservoir biogeochemistries and discuss implications for the efficiency of energy recovery.

## RESULTS

### The plasma membrane lipidomes of *H. congolense* WG10 and enrichment consortia are substantially altered by growth conditions and biofilm formation

*Halanaerobium congolense* WG10 was cultivated using a chemostat bioreactor vessel under three salinities (7%, 13%, and 20% NaCl) and operated in a continuous flow mode under three hydraulic retention times (19.2, 24, and 48 h). Planktonic growth was achieved by constant stirring at 50 rpm. The optimal growth salinity range for *H. congolense* is 10%–15% (29); therefore, in this study, we chose 13% NaCl as the optimal salinity, while 7% and 20% represent hyposaline and hypersaline conditions, respectively. Mixed microbial consortia in shale-produced fluids were enriched in chemostat bioreactors with no nutritional amendment under three hydraulic retention times (19.2, 24, and 72 h). In addition to the planktonic growth experiments, *H. congolense* WG10 was cultivated in a drip flow biofilm reactor equipped with silica coupons and fed with a defined saltwater liquid medium of three salinities (7%, 13%, and 20% NaCl). Likewise, mixed microbial consortia in shale-produced fluids were enriched in a drip flow bioreactor with no nutritional amendment. To couple laboratory experimentation with field evidence, produced fluids were collected from the gas–water separator of a shale well at four time points between December 2019 and May 2021 and filtered on-site. Microbial consortia retained on filter surfaces were flash-frozen, preserved on dry ice during transport, and stored at −80°C until analysis.

Lipids were extracted from all reactor-incubated and field-obtained microbial pellets and analyzed using liquid chromatography electrospray ionization tandem mass

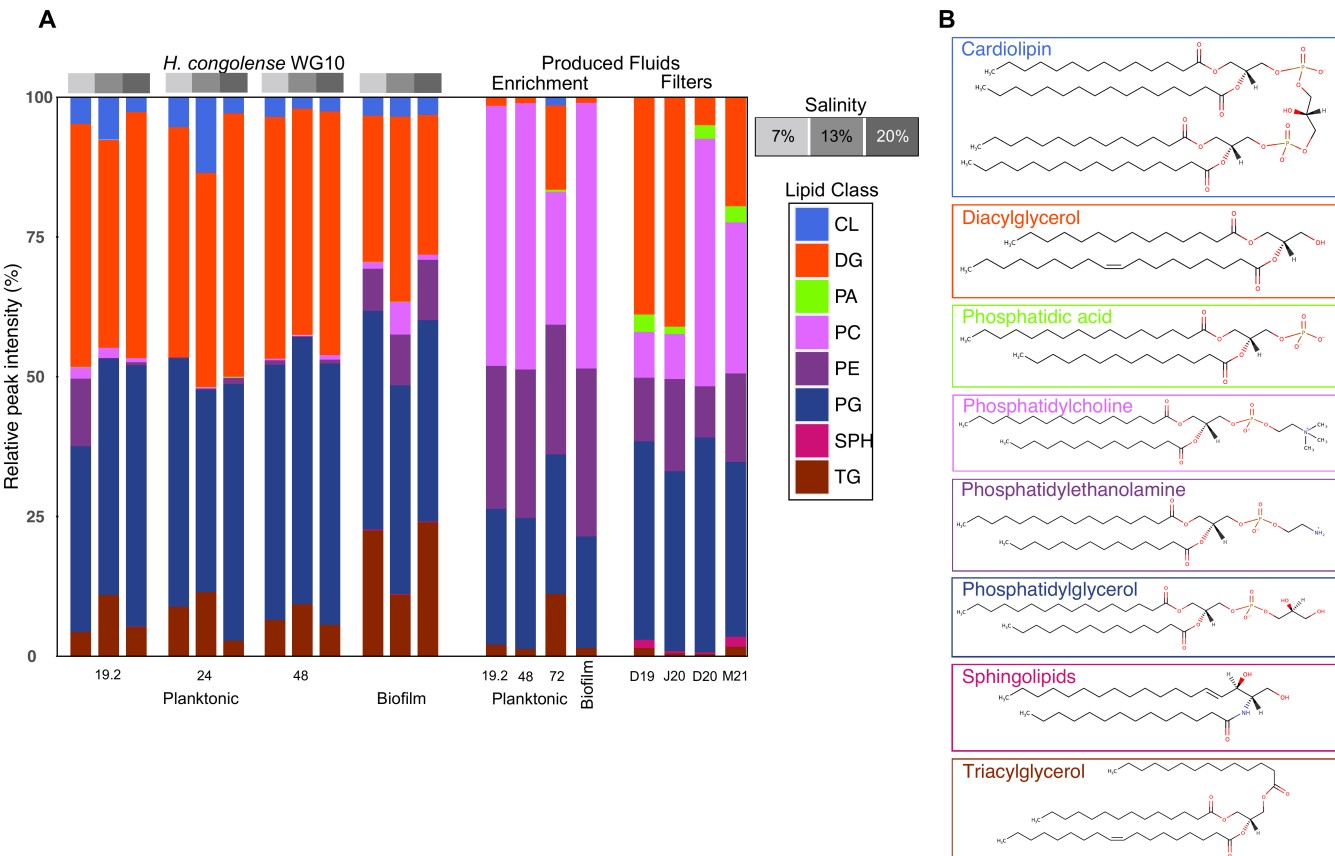

**FIG 1** (A) Relative abundances of lipid classes detected by LC-MS/MS in negative and positive ionization modes in *H. congolense* WG10, produced fluid enrichment cultures, and uncultivated field (filter) consortia. (B) Representative structures of each lipid class. Data from both ionization modes were first separately normalized by total intensity. S7, S13, and S20 represent salinity gradients (% NaCl), while 19.2, 24, 48, and 72 are hydraulic retention times in hours (h). HC, *Halanaerobium congolense*; PEL, planktonic; BF, biofilm; PFE, produced fluid enrichment; SPH, sphingolipids; D19, December 2019; J20, July 2020; D20, December 2020; M21, May 2021.

spectrometry (LC-ESI-MS/MS) in both positive and negative ionization modes. In total, we detected 194 lipids covering three categories (glycerophospholipids, sphingolipids, and glycerolipids) and 13 subclasses. Figure 1 shows the relative abundance of lipid classes detected across samples alongside representative structures. Lipids obtained in the negative ionization mode include PG, PE, phosphatidic acid (PA), and CL. PC, TG, DG, ceramide, and hexosylceramide (HexCer) were obtained in the positive mode. Overall, lipid metabolism in fractured shale microorganisms strongly varied by culture mode (biofilm vs planktonic) as well as growth conditions. *Halanaerobium congolense* WG10 cultures, regardless of salinity and HRT gradients, were more enriched in anionic phospholipids (PG and CL) compared to produced fluid enrichment and uncultivated field consortia, both of which had substantially higher amounts of zwitterionic phospholipids (PC and PE). This hints at an important role for zwitterionic membrane lipids in the community biology of the fractured shale microbiome. Clustering heatmaps of the 50 most discriminating lipid species collected in both positive and negative ionization modes are shown in SI Appendix, Fig. S2. Diacylglycerols were found in relatively high amounts in all the *H. congolense* WG10 samples as well as the filter consortia. This is unusual and discussed further in the SI Appendix.

To further explore multivariate patterns in the lipidomes of *H. congolense* WG10 culture samples, we performed unsupervised hierarchical clustering (Fig. 2), resolved by ionization modes. Based on lipids collected in the negative mode (PG, CL, PE, and PA), planktonic *H. congolense* WG10 samples grown under high salinity (20% NaCl)

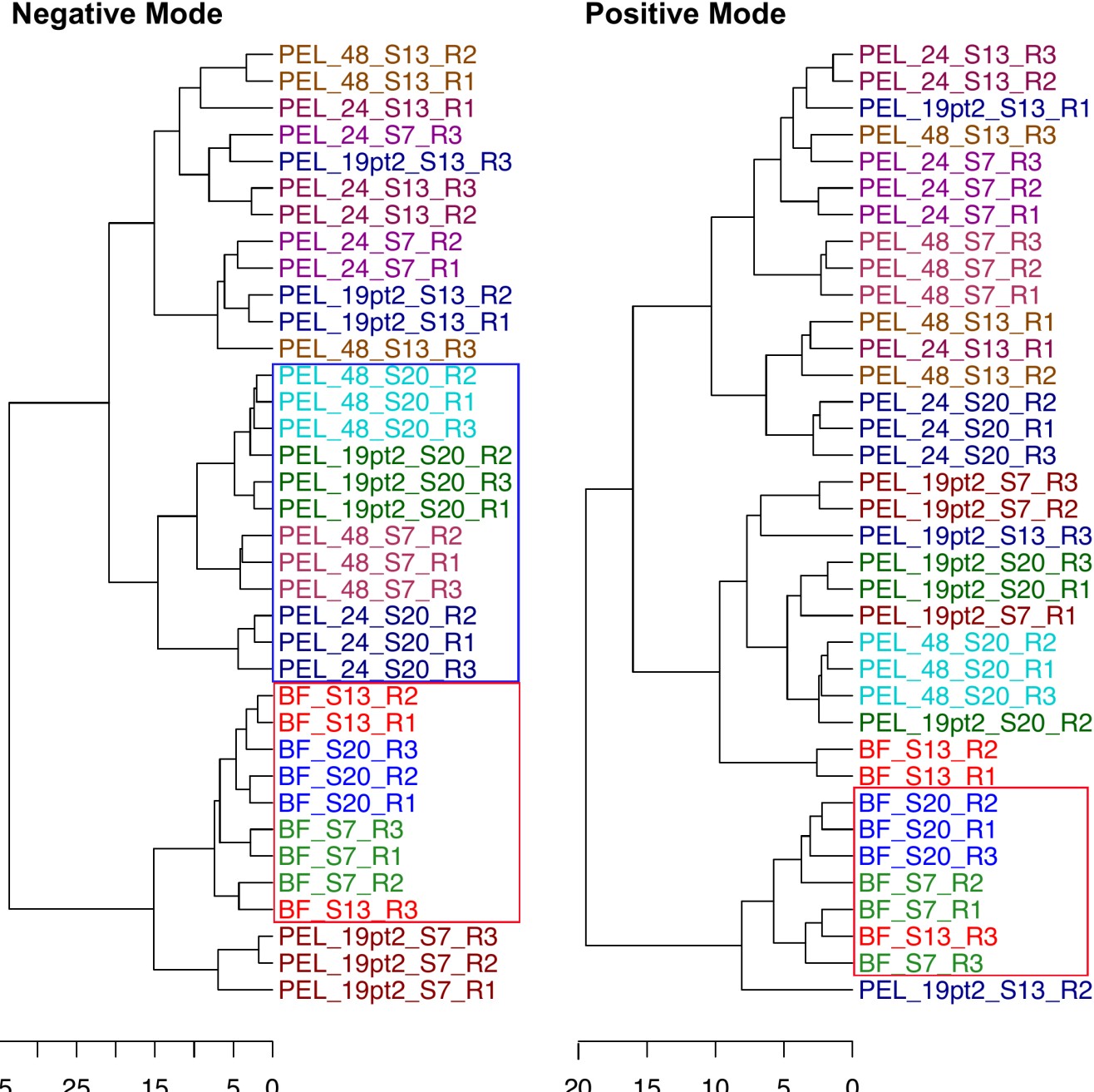

**FIG 2** Hierarchical clustering of *H. congolense* WG10 samples based on lipidomic data collected in the negative and positive ionization modes. S7, S13, and S20 represent salinity gradients (% NaCl), while 19.2, 24, and 48 are hydraulic retention times in hours (h). PEL, planktonic; BF, biofilm. The distance measure is Euclidean while clustering was done using the "Ward.D2" method in R, which implements Ward's 1963 criterion (30).

and variable HRT clustered together, with samples cultivated under low salinity plus high HRT (7% NaCl + 48 h). All *H. congolense* WG10 biofilm samples, irrespective of growth salinity, clustered together. With lipids obtained in the positive ionization mode, sample clustering by type was less pronounced; however, all *H. congolense* WG10 biofilm samples clustered together, except for two out of the three 13% NaCl replicates. These outcomes strongly suggest that adaptation to high salinity and biofilm formation in *H. congolense* WG10 requires patterned plasma membrane lipidome regulation. Further analyses were done to reveal these patterns. Structured membrane lipidome remodeling under various stresses is common in bacteria (14).

## Anionic PGs and CLs and zwitterionic PEs are important for *H. congolense* WG10 adaptation to high salinity

We assessed differences in the plasma membrane lipidomes of *H. congolense* WG10 grown under optimal salinity (13% NaCl) vs high salinity (20%) in both planktonic (at 24 h HRT) and biofilm modes. For the planktonic cultures, of the 194 lipids detected in both positive and negative ionization modes, only 18 (~9.2%) significantly differed based on Student's *t*-test ($P < 0.05$) and effect size [|log2 fold change (FC)| >1.5] criteria (Fig. 3; SI Appendix, Table S1). Out of these 18 differential lipids, 15 were elevated during growth at 20% salinity. These include phosphatidylethanolamine (4), phosphatidylglycerol (4), cardiolipin (3), and diacylglycerol (4). PE(16:0/18:1)/PE(16:1/18:0), PG(35:1), and PG(14:1/15:1) showed the highest log2 fold change increases—10, 9, and 8.5, respectively. The remaining three lipids, which were significantly reduced at 20% salinity, include phosphatidylethanolamine (2) and triacylglycerol (1). We report that ~47% of the lipids elevated at 20% salinity are anionic phospholipids (PGs and CLs) while ~27% are zwitterionic (PEs). No anionic lipid was suppressed under 20% salinity. Furthermore, we analyzed the fatty acid features of differentially expressed lipids in *H. congolense* WG10 planktonic cells growing under different salinities. The double bond index (DBI)

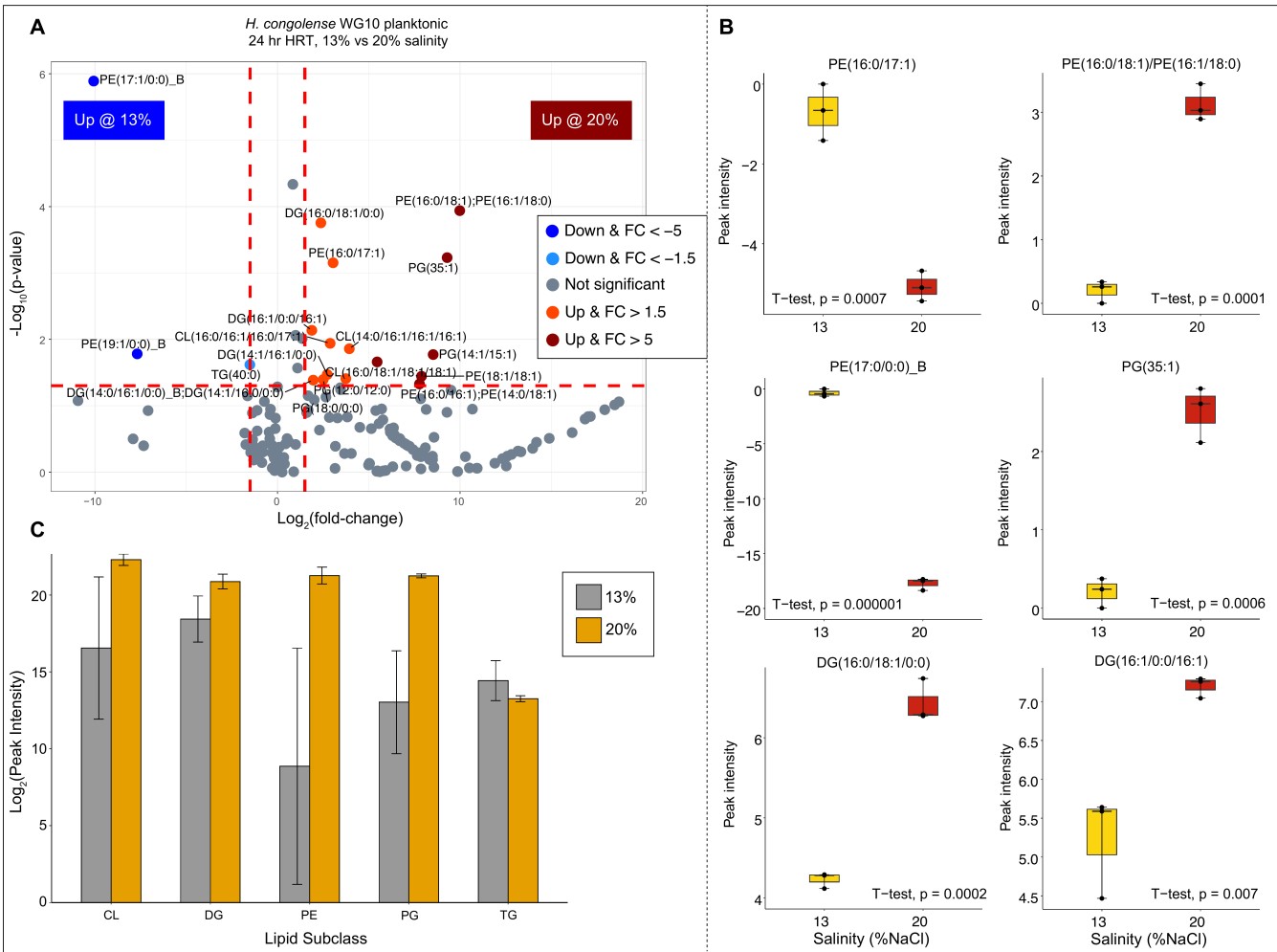

**FIG 3** Changes in abundance of membrane lipids in *H. congolense* WG10 planktonic cultures grown at 24 h HRT under 13% vs 20% NaCl. (A) Volcano plot showing the *P*-value and log2 fold change for each lipid. The direction of comparison is 13%–20%. The red horizontal line indicates the threshold of statistical significance, $P < 0.05$, while both red vertical lines delineate |FC| > 1.5. (B) Boxplots of the most significantly differential lipids ($P \leq 0.01$ and |FC| > 1.5). Peak intensities were log2-transformed then median-normalized, sample-wise. (C) Lipid variation by subclass using only the differential ($P \leq 0.05$ and |FC| > 1.5) species.

and mean chain length (MCL) were both higher at 20% compared to 13% (SI Appendix, Fig. S4A). This suggests increased bilayer fluidity and thickness, respectively, under high salinity compared to the optimum.

For *H. congolense* WG10 biofilm cultures, only 16 (~8.2%) out of all identified lipids differed significantly during growth at optimal salinity (13%) compared to high salinity (20%) (Fig. 4; SI Appendix, Table S2). Of these 16 differential lipid species, only two were significantly elevated at 20% salinity, both of which are isomers of lysophosphatidylglycerol—PG (18:1/0:0)_A and PG (18:1/0:0)_B. Seventy-nine percent of the remaining 14 lipids, which were suppressed at 20% salinity, belong to the phosphatidylcholine subclass. The rest include phosphatidic acid (14%) and phosphatidylglycerol (7%). The abundance of PC (18:0/19:1) is reduced by over 14-folds under 20% salinity. Similar to cultures in the planktonic mode, the only two lipids elevated at 20% salinity are anionic. Evidently, anionic phospholipids (PGs and CLs) and PEs help *H. congolense* WG10 cope with high salt stress. The overwhelming suppression of PCs in biofilm cultures growing under high (20%) salinity points to PC recycling to generate osmoprotecting choline moieties. In contrast to the planktonic cultures, there was an approximately fourfold decrease in both the MCL and DBI of the differential membrane lipids in *H. congolense* WG10 biofilm cells under 20% salinity relative to 13% (SI Appendix, Fig. S4B). This means

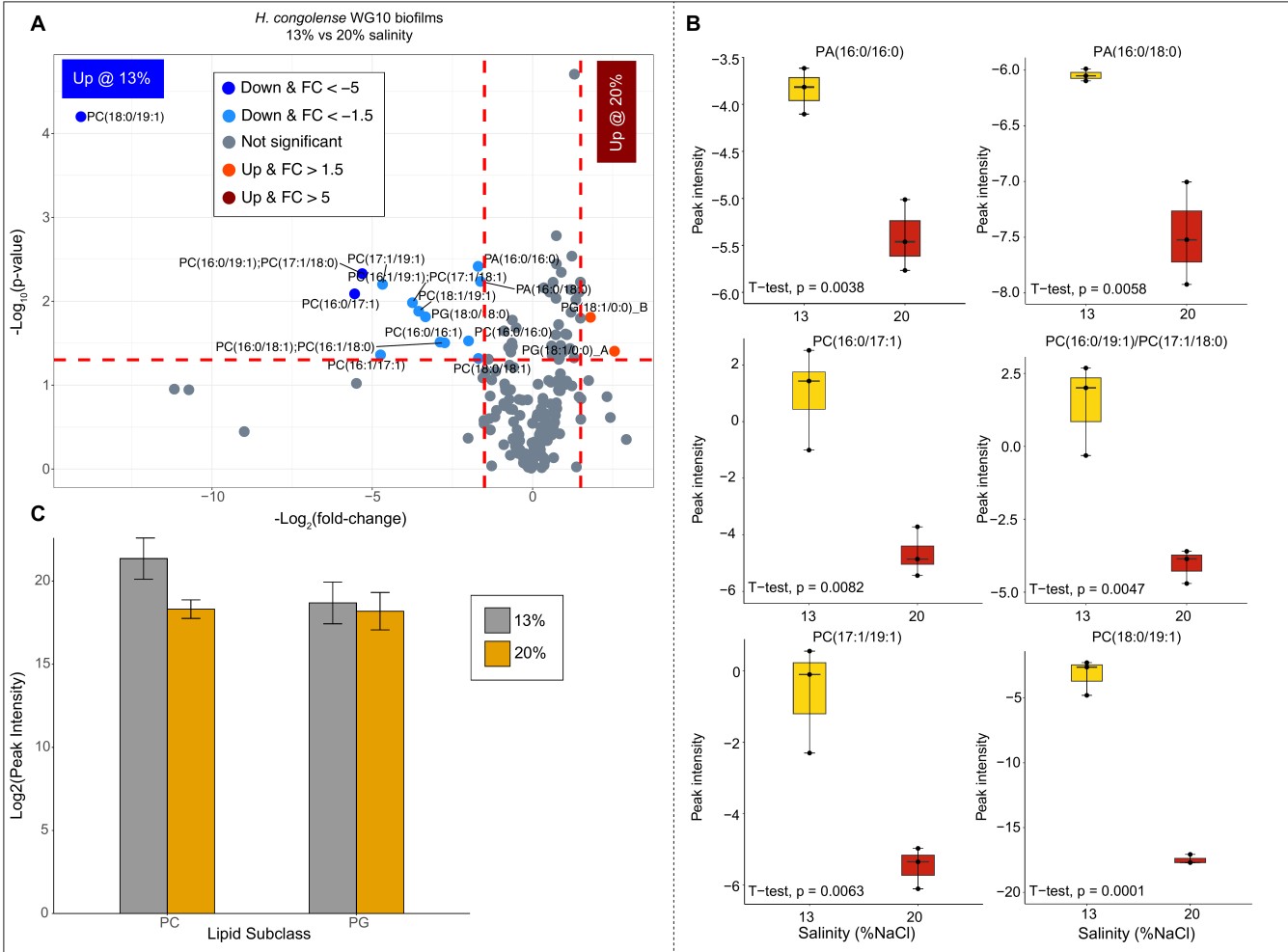

**FIG 4** Changes in abundance of membrane lipids in *H. congolense* WG10 biofilm cultures grown under 13% vs 20% NaCl. (A) Volcano plot showing the *P*-value and log2 fold change for each lipid. The direction of comparison is 13%–20%. The red horizontal line indicates the threshold of statistical significance, *P* < 0.05, while both red vertical lines delineate |FC| > 1.5. (B) Boxplots of the most significantly differential lipids (*P* ≤ 0.01 and |FC| > 1.5). Peak intensities were log2-transformed then median-normalized, sample-wise. (C) Lipid variation by subclass using only the differential (*P* ≤ 0.05 and |FC| > 1.5) species.

a decrease in both bilayer fluidity and thickness under high salinity compared to the optimum.

## Zwitterionic phospholipids modulate the biofilm physiology of fractured shale bacteria

We sought to interrogate the role of plasma membrane lipidome chemistry in the community behavior of shale reservoir bacteria. First, we compared the plasma membrane lipidome of *H. congolense* WG10 planktonic cells (grown at 48 h HRT) to the lipidome of its biofilm counterpart (incubated for 48 h). Salinity was maintained at the optimum (13%) for both growth modes. Based on statistical significance ($P < 0.05$) and effect size ($|FC| > 1.5$) criteria, 76 (~39%) out of all 194 identified lipids differentially varied (Fig. 5A; SI Appendix, Table S3). Only 2 out of these 76 species, HexCer (d18:2/16:1)_B and PG (12:0/13:0), were suppressed in *H. congolense* biofilms relative to planktonic cells. The remaining 74 (97%) increased in abundance during biofilm growth, and they include a broad range of lipid classes.

Next, we adjusted the statistical significance and effect size thresholds to $P < 0.01$ and $|FC| > 2.5$, respectively, to narrow down on the most discriminant lipids. Only 26 lipids met these criteria, among which 12 showed at least a 10-fold increase in *H. congolense* WG10 growing as biofilms relative to planktonic. Nine out of these twelve are phosphatidylcholines, two are phosphatidylethanolamines, and the remaining one is a cardiolipin. The dominance of PCs and PEs among lipids most significantly elevated in biofilms vs planktonic indicates that zwitterionic lipids may modulate the community physiology of *H. congolense* WG10, which ubiquitously persists in fractured shale reservoirs. Upon analysis of the fatty acid chemistry of the differential lipids, we found a higher mean chain length and double bond index during biofilm growth compared to planktonic (SI Appendix, Fig. S4C). This suggests that an increase in plasma membrane fluidity and thickness is important for *H. congolense* WG10's biofilm physiology.

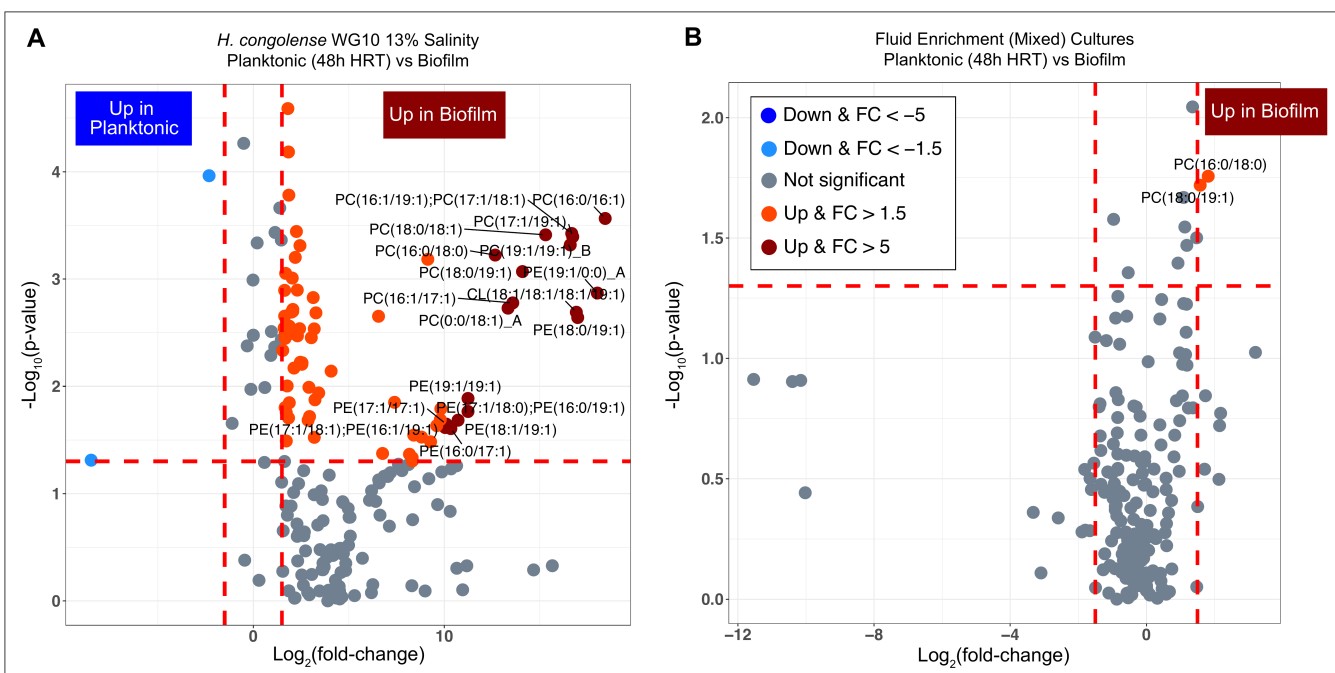

**FIG 5** Changes in abundance of membrane lipids in fractured shale microbes during biofilm growth vs planktonic. (A) Volcano plot showing the *P*-value and log2 fold change of lipids in *H. congolense* WG10 growing as biofilms (incubated for 48 h) vs planktonic (grown under 48 h HRT). Salinity was maintained at the optimum (13%) for both growth modes. (B) Volcano plot showing the *P*-value and FC of lipids in mixed microbial consortia in produced fluids enriched as biofilms (incubated for 48 h) vs planktonic (grown under 48 h HRT). The direction of comparison is planktonic to biofilm. The red horizontal line indicates the threshold of statistical significance, $P < 0.05$, while both red vertical lines delineate $|FC| > 1.5$.

Furthermore, we compared the membrane lipidome of mixed microbial consortia in shale-produced fluids enriched as biofilms (incubated for 48 h) to that of its planktonic counterpart (grown under 48 h HRT). Only 2 of the 194 detected lipids—PC (16:0/18:0) and PC (18:0/19:1)—were significantly differential ($P < 0.05$ and $|FC| > 1.5$) (Fig. 5B; SI Appendix, Table S4). Both of them, zwitterionic phosphatidylcholines, were elevated during biofilm growth. This supports our notion that zwitterionic plasma membrane lipids are important to the community behavior of the fractured shale microbiome.

## Lipid signatures can predict *H. congolense* WG10 biofilm growth

We performed a multivariate receiver operator characteristic (ROC) curve-based exploratory analysis using the Biomarker Analysis module of MetaboAnalyst v5.0 (31) to predictively model the differences between the lipidomes of *Halanaerobium congolense* WG10 biofilm and its planktonic counterpart. Implementing random forests (RFs) as the classification and feature ranking method, we built and evaluated machine learning models using all 194 identified lipids found in the 36 combined samples of *H. congolense* WG10 planktonic ($n = 27$) and biofilm ($n = 9$) cultures. A panel of 15 lipids (Fig. 6A) resulted in an area under the curve value of 0.997 (SI Appendix, Fig. 2A) and predictive accuracy of 93.8% (SI Appendix, Fig. 2B). They include PE(17:1/18:0)/PE(16:0/19:1), DG(32:1), PE(16:0/17:1), PE(19:1/19:1), DG(16:1/0:0/16:1), PE(17:1/19:1), PC(16:1/0:0)_B, HexCer(d18:2/16:1)_B, CL(17:1/16:0/16:1/17:1), DG(14:0/16:1/0:0)_B/DG(14:1/16:0/0:0), DG(15:1/16:1/0:0), DG(14:0/16:1/0:0)_A, PG(14:0/0:0)_B, CL(14:0/16:1/16:1/16:1), and PG(16:0/16:1)_B, arranged in the order of decreasing importance. Among these lipids, only the PEs (all four of them) were found to be elevated in the biofilms, a further indication of the role of zwitterionic lipids in *H. congolense* WG10 biofilm (community) biology. Partial least squares discriminant analysis (PLS-DA) with a two-latent variable input supported the RF modeling outcome, reliably separating (accuracy = 0.97, $R^2$ = 0.89, $Q^2 = 0.81$) *H. congolense* WG10 samples into biofilms and planktons (Fig. 6B).

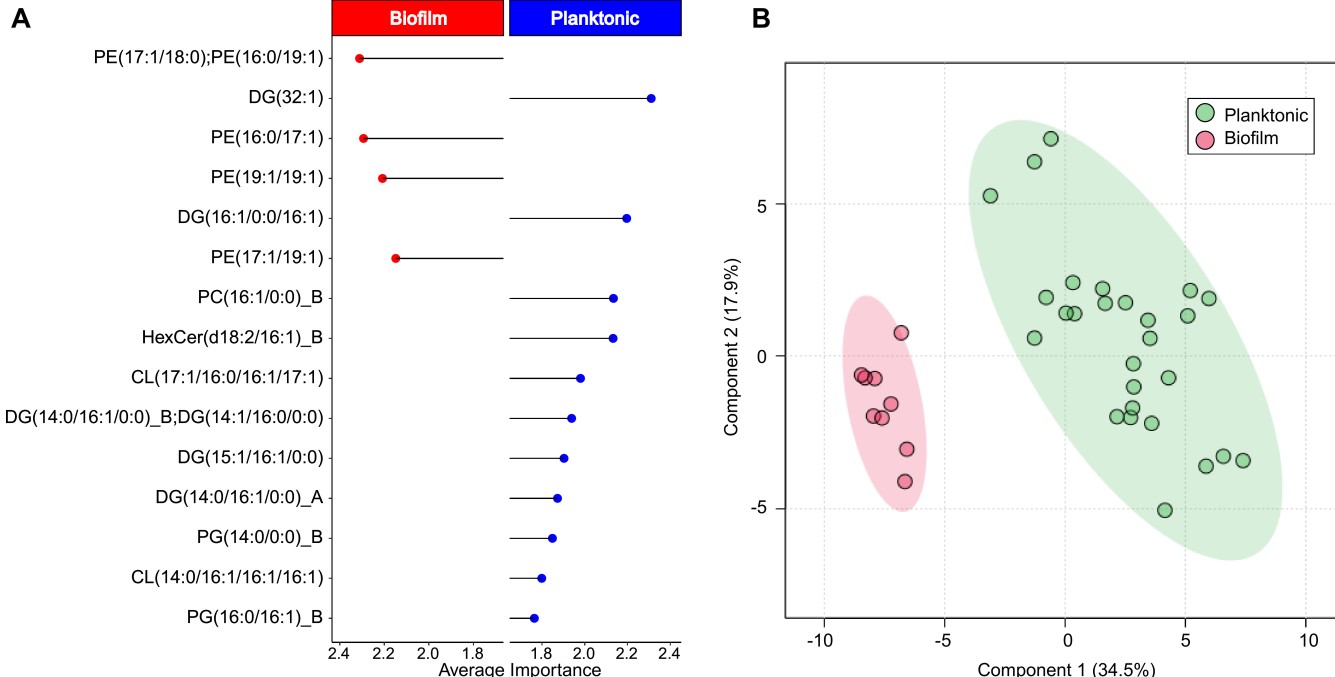

**FIG 6** Machine learning-based predictive modeling of *H. congolense* WG10 biofilm growth. (A) Random forest classification performance of the 15 most discriminant lipids ranked by mean importance measure. Red and blue denote higher abundance in biofilm and planktonic cultures, respectively. (B) Partial least squares discriminant analysis two-dimensional score plot showing a distinct separation of *H. congolense* WG10 culture samples into biofilms and planktons.

# Fractured shale bacteria predominantly increase their membrane anionic PG composition under longer residence times

We compared the lipidomes of *Halanaerobium congolense* WG10 planktonic cultures grown at optimal salinity (13% NaCl) under 19.2 h (lowest) vs 48 h (highest) HRT. Lipid species were considered significantly differential if they met both of these criteria: *P*-value <0.05 and |FC| > 1.5. Out of the 194 detected lipids, only 13 (6.7%) were significantly variant. These include phosphatidylglycerol (7), cardiolipin (3), diacylglycerol (2) and phosphatidylethanolamine (1) (Fig. 7A; SI Appendix, Table S5). Notably, all six lipids that increased under 48 h HRT relative to 19.2 h HRT are PGs. Only one PG, PG(14:1/14:1), alongside the remaining six differential lipids (CLs, DGs, and a PE), was found at higher abundance under 19.2 h HRT. The mean chain length and double bond index of the differential lipids in *H. congolense* WG10 growing under 48 h HRT were only slightly less than that at 19.2 h (SI Appendix, Fig. S4D).

We performed a similar comparative analysis for mixed microbial consortia in shale-produced fluids enriched in the planktonic mode under 19.2 h HRT (lowest) vs 72 h HRT (highest). An overwhelming 100 (~52%) out of 194 detected lipids, covering all subclasses, met the statistical significance (*P* < 0.05) and effect size (|FC| > 1.5) benchmarks (Fig. 7B; SI Appendix, Table S6). Thirty-one of these differential species were increased under 72 h HRT relative to 19.2 h HRT: 16 of them (51.6%) are PGs, 9 are DGs, 3 are CLs, 2 are PAs, and 1 is a TG. Remarkably, for both *H. congolense* WG10 isolate cultures and the mixed consortia, no zwitterionic phospholipid was significantly increased under the highest HRT (48 and 72 h, respectively) compared to the least (19.2 h). Phosphatidylglycerol species dominated the panel of lipids found to be elevated under the highest HRT. Similar to the pure culture, differential lipids in the mixed consortia had a slightly lower MCL and DBI at 72 h HRT compared to 19.2 h (SI Appendix, Fig. S4E).

## DISCUSSION

Anionic PGs and CLs and zwitterionic PEs (Fig. 3 and 4) predominated lipids that significantly increased in *H. congolense* WG10 planktonic and biofilm cultures under high salinity (20% NaCl) relative to the optimum (13% NaCl).

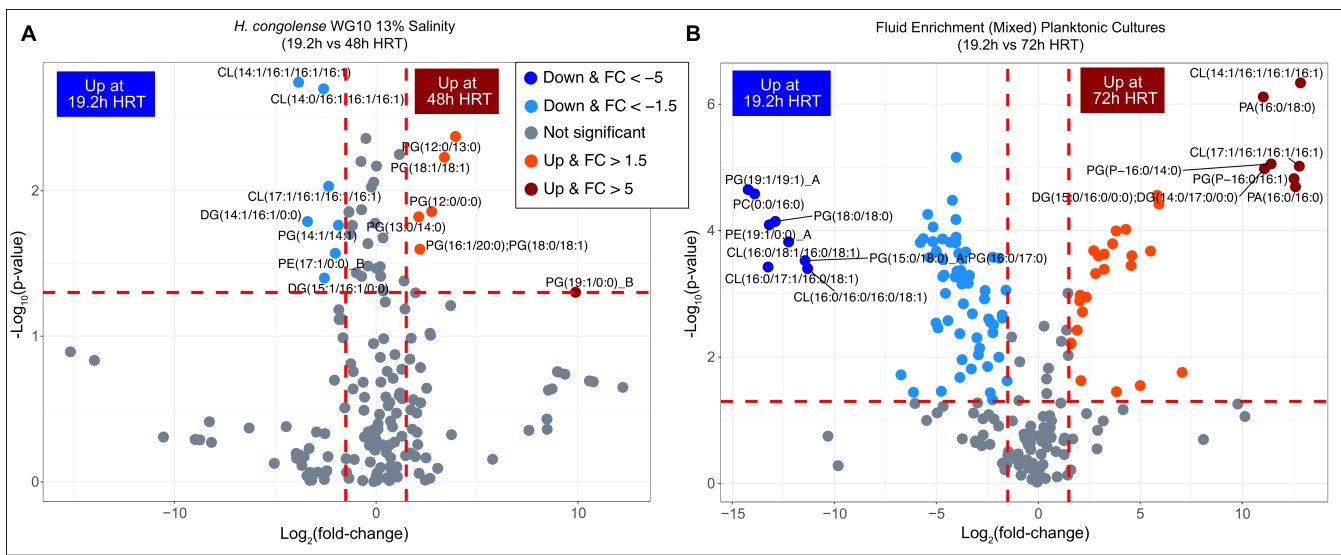

**FIG 7** Comparison between the lipidomes of fractured shale microorganisms grown under low vs high hydraulic retention times. (A) Volcano plot showing the *P*-value and log2 fold change of lipids in *H. congolense* WG10 planktonic cultures grown at optimal salinity (13% NaCl) under 19.2 h (lowest) vs 48 h (highest) HRT. (B) Volcano plot showing the *P*-value and FC of lipids in mixed microbial consortia in shale-produced fluids enriched in planktonic modes under 19.2 h (lowest) vs 72 h (highest) HRT. The direction of comparison is 19.2 h to 48 or 72 h. The red horizontal line indicates the threshold of statistical significance, *P* < 0.05, while both red vertical lines delineate |FC| > 1.5.

The *H. congolense* genome encodes phosphatidylglycerophosphatase A (*pgpA;* accession ID: SAMN04515653_10658) and cardiolipin synthase (*cls,* accession ID: SAMN04515653_11647), which catalyze PG and CL synthesis, respectively. Hypersalinity exerts osmotic stress and ionic toxicity on biological cells (32, 33), causing loss of intracellular water via exosmosis and direct damage to cellular structures including the plasma membrane. Monovalent cations, such as Na$^+$, induce hydrophobic interactions among membrane lipid headgroups, which result in biophysical transition from lamellar (normal bilayer configuration) to deleterious non-lamellar (hexagonal-II or cubic) phases (34, 35). An increase in levels of anionic PGs and CLs, which we observed in *H. congolense* WG10 growing under high salinity, is necessary to stabilize the membrane in the normal bilayer (lamellar) orientation. This is because unlike their zwitterionic counterpart, anionic headgroups are not very susceptible to such cation-induced interactions (27, 34, 36, 37). Moreover, anionic PLs are thought to play essential roles in osmosensing, accumulation of ions at the cell–environment interface by electrostatic forces, and channel activation for osmolyte acquisition (23, 24, 38). They are, therefore, key to bacterial survival in high-salinity shale reservoirs, through stabilizing the membrane's normal bilayer phase and enabling "salt-in" osmoadaptive functions. Prior studies support the role of anionic phospholipids in bacterial hypersalinity tolerance (23, 39, 40). We are not sure why there was a net increase in membrane PE composition in *H. congolense* WG10 planktonic cells under high salinity as prior studies have established an opposite trend (24, 41). However, being the major phospholipid subclass in most bacterial membranes (42), they might be key to maintaining the default molecular order of the matrix and modulating its adaptive biophysical modifications and functions. It is important to take into consideration the difficulty of directly comparing the magnitude of lipidomic changes induced by each growth parameter.

Phosphatidylcholines were dramatically suppressed in *H. congolense* WG10 biofilm cells growing under high salinity relative to the optimum (Fig. 4). This suggests subsequent PC degradation to choline residues. We believe that like several bacteria (43–45), *H. congolense* WG10 expresses functional phospholipases that hydrolyze redundant PCs to choline (see Supplementary Appendix Fig. S1 and text). Phospholipase is, in fact, encoded in the genome of a *H. congolense* strain (accession ID: SAMN04515653_13217). Therefore, it is possible that this bacterium accumulates choline as an osmoprotectant (10) in a controlled laboratory reactor with limited alternatives. Proteomics and metabolomics are needed to verify this. However, within the fractured shale ecosystem, *H. congolense* might utilize other osmoprotecting compounds such as sugars and amino acids (10), therefore dispose choline (if accumulated), which it lacks the ability to catabolize. Other shale-dominating taxa including Halomonadaceae and *Marinobacter* can scavenge and oxidize choline to glycine betaine (3), which serves as a nutrient/energy source, as well as an osmoprotectant for several species including *Halanaerobium*. Viral infection and lysis of GB producers most likely account for the supply of GB to non-producers (3). In a prior study, GB levels in shale-produced fluids were positively correlated with well salinity (10). We argue that co-metabolic PC hydrolysis in osmotically stressed *Halanaerobium* and choline oxidation in other taxa are the missing intermediate pieces of this puzzle. Our finding adds to growing evidence that *H. congolense* utilizes the "compatible solutes" (choline and possibly GB) strategy for high salinity stress adaptation, in addition to the more established salt-in mechanism (10, 25). This, however, needs to be further substantiated by proteomics and metabolomics analyses.

While there are strong indications that in engineered shale, *Halanaerobium* accumulates GB as an osmoprotectant, some strains might ferment it to trimethylamine (3), which is readily consumed by methanogens especially *Methanohalophilius* to produce methane (3, 10). The enzyme responsible for fermentative GB reduction via the Stickland reaction, GB reductase (*grdHI*), is encoded in some shale *Halanaerobium* genome (3). Also see the accession ID: SAMN04515653_10521. Biogenic methanogenesis enhances secondary natural gas recovery from reservoirs (46). By implication, *H. congolense* adaptation to high salinity through PC suppression and subsequent recycling could be

intricately linked to increased biogenic methane production. The fact that this phenomenon was only observed in the biofilm cells suggests that execution of the choline-based "compatible solutes" strategy has unfavorable energetics and is largely incumbent on PC overabundance and constraints to extracellular ion acquisition, which might be imposed by the biofilm architecture. In fact, *H. congolense* planktonic cultures showed relatively marginal PC abundance compared to their biofilm counterpart (Fig. 1).

As fractured shale wells mature, reservoir salinity increases (18), and so does fluid retention time. The rate of flowback and produced water recovery fluctuates—it naturally declines over time due to constant production and is deliberately controlled in line with seasonal demand (47). We simulated this phenomenon in the lab using the hydraulic retention time concept in a continuous culture setup. Moreover, studying cells at a steady state is the only way to obtain meaningful and reproducible results (48). The longer the medium/fluids spend in the shale reservoir or at the lab scale, in the chemostat bioreactors, the more resident bacterial cells are metabolically stressed due to nutrient (carbon) depletion and accumulation of toxic metabolic by-products. Anionic phosphatidylglycerols predominated lipid species that significantly increased in both *H. congolense* WG10 cells and mixed microbial consortia, grown and enriched, respectively, under the highest HRT (48 and 72 h, respectively) (Fig. 7) relative to the lowest (19.2 h). The fact that no zwitterionic lipid was significantly increased under the highest HRTs in both the single and mixed cultures suggests that the upsurge in PGs might have been geared in large measure toward increasing the net negative charge of the membrane and cell surface. This perhaps helps repel toxic anionic metabolites such as sulfide and organic acids (49) from the cell envelope. In addition, PGs facilitate protein folding and binding (50) in bacteria, processes that are key to cellular survival under metabolic stress.

Biofilms in engineered shale reduce rock permeability and clog fracture networks that constitute hydrocarbon flow channels (8, 51). They also confer increased ecological persistence and biocide resistance on the inhabitant microbiome. The plasma membrane directly mediates cellular surface attachment and biofilm development, and this depends on the bilayer's physicochemical attributes, largely determined by its intact polar lipid headgroup chemistry (52). In this study, we found that zwitterionic phosphatidylcholines and phosphatidylethanolamines overwhelmingly predominated lipids elevated in *H. congolense* WG10 and mixed microbial consortia enriched from shale-produced fluids during biofilm growth relative to planktonic (Fig. 5). Moreover, in uncultivated microbial consortia obtained from field-filtered shale-produced fluids sampled across four time points, the relative abundance of zwitterionic PCs and PEs spiked in December 2020 (Fig. 8) following a steep drop in well flow rate 5 months prior, corresponding to an increase in HRT. This is offset by a decrease in levels of neutral lipids. Anionic lipids are maintained at fairly high proportions due to their ever-needed role in high salt tolerance. Low shale well flow rates would typically facilitate cellular attachment and biofilm formation in the reservoir.

Hence, our hypothesis is that PCs and PEs modulate the community (biofilm) physiology of the fractured shale microbiome. Their roles might include decreasing energy-intensive membrane fluidity (53) [an unnecessary burden to sessile biofilm cells with attenuated metabolic activity (54)], reducing the net negative charge of the cell envelope to prevent cell–cell (or cell–surface) electrostatic repulsion, and forming specialized lipid rafts to present protein receptors for quorum sensing and other intercellular exchanges. A prior study reported that levels of PCs and PEs were higher in the early and mature phases of biofilm formation in *Candida albicans* compared to the planktonic cells (52). This opens up a new frontier of discussion about the use of choline-containing additives for shale well development, such as choline chloride currently preferred as clay stabilizers to the more toxic tetramethylammonium chloride (55). Free choline residues could be acquired by shale microbes such as *Marinobacter* (3) and assimilated into membrane PCs, which we believe facilitate deleterious biofilm formation. Geophysical and technical constraints make it nearly impossible to sample and monitor intact biofilms in fractured shale reservoirs. Our machine learning model based

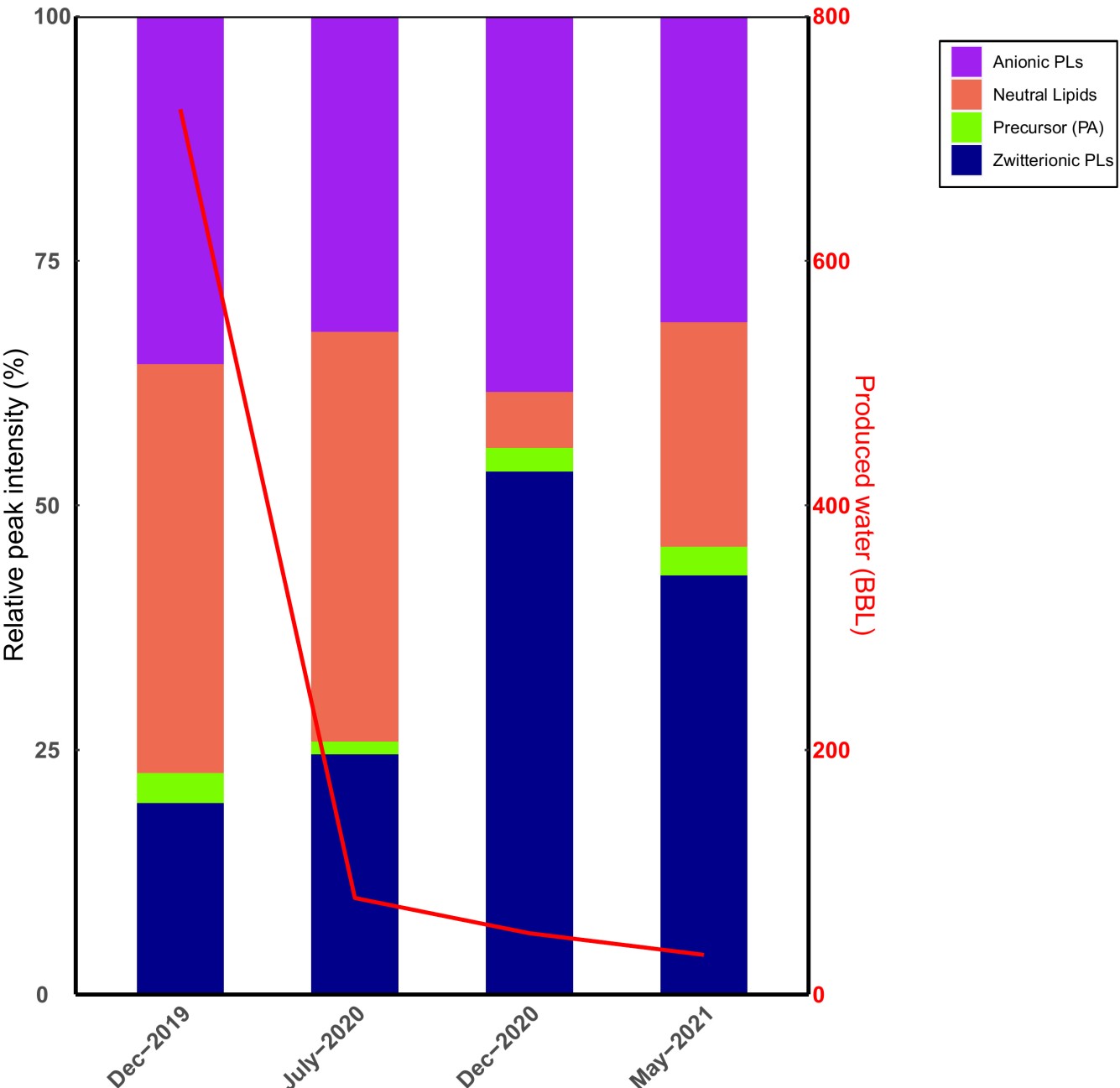

**FIG 8** Temporal variations in the net charge of lipids found in mixed microbial consortia from field-filtered shale-produced fluids sampled across four time points. The stacked bars are overlaid with a line graph showing corresponding volumes of produced water recovered from the well. Anionic = sum of phosphatidylglycerols and cardiolipins; zwitterionic = sum of phosphatidylcholines and phosphatidylethanolamines; neutral = sum of diacylglycerols, triacyclglycerols, and sphingolipids.

on 15 discriminant lipids (Fig. 6) is a useful tool for estimating biofilm growth in non-sterile subsurface hydrocarbon systems using lipid signatures in produced fluids. A sudden increase in PE abundance could especially be indicative of aggressive biofilm development.

Besides salinity, hydraulic retention time, and biofilm formation, whose effects were evaluated in the present study, there are other strong drivers of plasma membrane lipid chemistry adjustments in fractured shale microbes, including temperature and pressure. The downhole temperature and pressure of fractured shale reservoirs vary

depending on formation characteristics and operating parameters and fluctuate over time during and after well development (56, 57). This drives adaptive morphological and physiological remodeling in the inhabitant microbiome. High temperature and low pressure exert a similar effect on the biophysical properties of the membrane, which is a disruption of acyl chain packing order, which results in an increase in fluidity (58, 59). In response, bacteria primarily increase the proportion of saturated fatty acids in the bilayer. Conversely, low temperature and high pressure gelatinize the membrane, necessitating an increase in levels of unsaturated and branched chain fatty acids to restore optimal fluidity. This phenomenon is commonly known as homeoviscous adaptation (60). In addition, temperature and pressure changes affect headgroup chemistry. In general, high temperature and pressure usually induce an overall increase in relative abundance of polar lipids in the membrane (59, 61). Also, zwitterionic lipids, which possess balanced positive and negative charges, pack more densely than their anionic counterparts, hence are elevated in some bacteria under high temperature and low pressure (59). It is, thus, important to keep in mind that it is an interplay of these factors—including temperature, pressure, salinity, and hydraulic retention time—that will determine the bulk plasma membrane biophysical properties of the fractured shale microbiome.

Microbial plasma membrane intact polar lipid chemistry adjustments in response to environmental and engineered changes in fractured shale reservoirs have significant implications for biocontrol efficacy. Non-oxidizing glutaraldehyde (GTD) and quaternary ammonium compounds (QACs) are biocides very commonly used to control microbial activities in engineered wells (28). QACs either directly target the plasma membrane or use it as a conduit to access the intracellular space. On the other hand, GTD interacts with unprotonated amines on the cell surface, which disrupts membrane and cell wall integrity (62, 63). QACs, which possess a cationic headgroup, electrostatically bind the cell envelope, inserting their hydrocarbon tails into the matrix in ways that cause lethal structural disruptions (64).

Our results indicate that even though anionic phospholipids are elevated under increasing salinities (Fig. 3 and 4), levels of zwitterionic species increase dramatically during controlled biofilm formation (Fig. 5) and peak in shale-produced fluids immediately following a steep drop in flow rates (Fig. 8). Zwitterionic lipid headgroups have a perfect balance of positive and negative charges, therefore reducing the zeta potential of the membrane. This might limit the efficacy of cationic biocides such as QACs—used in nearly every shale formation in the US (28)—whose action relies on electrostatic binding to the cell surface. However, biocidal efficacy non-linearly depends on various other factors such as the salinity of the formation. At high salinity, the reactive intermediate of GA is stabilized, potentially making it less reactive and effective (65). However, high salinity enhances the hydrophobicity of the membrane bilayer (66), which might facilitate the intercalation of the carbon tails of QACs into the matrix, therefore increasing biocidal efficacy. Further experiments under controlled conditions are needed to investigate how induced plasma membrane lipidomic changes in fractured shale taxa affect biocidal efficacy. Overall, bacterial communities may better thrive in shale reservoirs under lower (within the optimal range) salinity and flow rates due to increased plasma membrane zwitterionic phospholipid composition, which facilitates biofilm formation and might increase resistance to cationic biocides.

## Conclusion

Analyzing the lipidomes of *H. congolense* WG10 and mixed consortia in shale-produced fluids revealed patterned plasma membrane chemistry and biophysical adjustments underlying adaptations to high salinity and hydraulic retention time. This study has also elucidated the intricacies of membrane remodeling in shale taxa that support biofilm growth. While anionic phospholipids might help the shale microbiome survive high salt and metabolic stresses, zwitterionic species predominate their plasma membrane lipidome during biofilm formation. These adaptive membrane physicochemical modifications have ecosystem-scale implications for carbon cycling, biogenic

methanogenesis, biofilm formation, microbial persistence, and biocide resistance in engineered shale. This study contends that plasma membrane dynamics hold enormous insights into microbial activities in natural and disturbed environmental systems such as subsurface hydrocarbon reservoirs, which could inform more effective monitoring, biocontrol, and beneficial exploitation.

## MATERIALS AND METHODS

### Field sampling

Produced fluids were sampled four times (December 2019, July 2020, December 2020, and May 2021) from the gas–water separator of a hydraulically fractured natural gas well in the Appalachian Basin (Marcellus Shale Energy and Environmental Laboratory—MSEEL II Boggess wells in West Virginia). Portions of the raw fluids were immediately stored in 1-L sterile amber glass containers. The remaining fractions were filtered using 0.45-µm PES filters (EMD Millipore, Burlington, MA, US) and with 0.22-µm mixed cellulose ester filters (90 mm diameter) (Millipore-Sigma, Burlington, MA). Filters (0.22 µm) containing bacterial residues were stored in sterile Ziploc bags, and the filtrates were retained in sterile glass containers. All samples were transported back to the laboratory on dry ice and preserved at −80°C until use.

### Planktonic growth experiments

We obtained the model organism *Halanaerobium congolense* WG10 (NCBI Assembly accession number: GCA_900102605.1) previously isolated from produced fluids sampled from a Utica/Point Pleasant natural gas well in Ohio, US (7). Isolates were grown at 40°C in chemostat bioreactors (Sartorius Biostat Q-plus, Germany) containing a defined saltwater medium of three salinities (7%, 13%, and 20% NaCl) and operated in a continuous flow mode under three hydraulic retention times (19.2, 24, and 48 h). HRT is conceptually the average length of time the medium, cells, and their products spend in the reactor before flowing out under the continuous culture setup. To vary the HRT, after the incubation experiment began with a fixed reactor volume, we adjusted the flow rate of the peristaltic pump. A 19.2 h HRT means that it would take 19.2 h for the original volume of the reactor to be completely replaced with fresh media (or completely depleted, if no continuous replenishment occurred). The medium contained the following: 1 g $L^{-1}$ $NH_4Cl$, 10 g $L^{-1}$ $MgCl_2 \bullet 6H_2O$, 0.1 g $L^{-1}$ $CaCl_2 \bullet 2H_2O$, 1 g $L^{-1}$ KCl, 0.5 g $L^{-1}$ cysteine, 1% Wolfe trace element solution, and 1% Wolfe vitamin solution. After autoclaving, it was amended with 10 mM D-glucose, 0.2% $NaHCO_3$ solution, and 0.003% phosphate solution ($KH_2PO_4$ and $K_2HPO_4$). Anaerobic conditions were maintained by constantly bubbling a mixture of high-purity $N_2$ and $CO_2$ into the vessel. The pH was maintained at 7.0 by the automatic addition of acid (1M HCl) and base (1M KOH). In a different setup, unfiltered produced fluids were enriched in chemostat bioreactors at 40°C under three HRTs (19.2, 48, and 72 h). For both setups, the density ($OD_{600 nm}$) of the culture medium was monitored daily until a steady state was attained (5–8 days). Afterward, cells were recovered by centrifugation at 4,000 × *g* for 30 min and frozen at −80°C before transport to the Pacific Northwest National Laboratory (PNNL), Washington, US, on dry ice.

### Biofilm growth experiments

Biofilm growth experiments were conducted using a continuous drip flow bioreactor (Biosurface Technologies, Bozeman, MT) designed for growth under low shear conditions. The system consisted of six parallel test channels, each holding one silica coupon the size of a standard microscopic slide (75 × 25 × 1 mm). In one setup, the silica coupons were inoculated with *H. congolense* WG10; then, the reactor was fed with defined saltwater medium of three salinities (7%, 13%, and 20% NaCl) and 25 mM glucose at a constant dilution rate (0.35 mL/min). In a different setup, mixed microbial consortia in shale-produced fluids were enriched in a similar biofilm reactor with no

nutrient amendment. For both setups, after an approximately 48-h incubation, each coupon containing a developed biofilm was removed from the reactor and immersed into a 50-mL Falcon tube containing 10-mL phosphate-buffered saline (PBS). The tube was vortexed to dislodge the biofilm into the solution; then, the coupon was removed. The cell suspension was centrifuged, and the resulting pellet was frozen at −80°C before transport to PNNL on dry ice.

## Lipid extraction

Lipid extraction was conducted at PNNL following the metabolite, protein, and lipid extraction (MPLEx) procedure (67). Cell pellets recovered from prior planktonic and biofilm growth experiments were washed with PBS and transferred to a MμlTI SafeSeal Sorenson microcentrifuge tube containing a 2:1:0.75 cold mixture of chloroform:metha-nol:water. An internal standard cocktail (10 μL, Avanti EquiSPLASH) was added, and the tube was vortexed for 10 s, then allowed to sit for 1 h at 4°C. It was centrifuged to separate extracts into three layers—a top hydrophilic layer containing polar metabolites, a protein interphase, and a bottom organic layer containing lipids. The organic phase was transferred to a new tube. The aqueous phase was re-extracted by adding 2 mL of fresh chloroform to it. The mixture was vortexed, allowed to sit at 4°C for 1 h, and centrifuged. The resulting bottom layer was added to the first organic phase. The pooled chloroform layers were evaporated to dryness under a stream of pure nitrogen and stored in 2:1 chloroform:methanol at −80°C until analysis.

## Mass spectrometry analysis

Total lipid extracts were dried again *in vacuo*, redissolved in chloroform (5 μL), and brought to a final volume of 50 μL with methanol. They were analyzed as previously described (67) by LC-ESI-MS/MS using a Waters Acquity UPLC H class system (Waters Corp., Milford, MA) interfaced with a Velos Elite Orbitrap mass spectrometer. Briefly, a 10-μL sample aliquot was injected onto a Waters Charged Surface Hybrid column (3.0 mm i.d. × 150 mm × 1.7 μm particle size) maintained at 42°C. Separation was done over a 34-min gradient using mobile phases of different polarities [A: ACN/H$_2$O (40:60) containing 10 mM ammonium acetate and B: ACN/IPA (10:90) containing 10 mM ammonium acetate] at a flow rate of 250 μL/min. The full gradient profile was as follows (min, %B): 0, 40; 2, 50; 3, 60; 12, 70; 15, 75; 17, 78; 19, 85; 22, 92; 25, 99; and 34,99. The UPLC system used a Thermo HESI source coupled to the mass spectrometer inlet. The MS inlet and HESI source were both maintained at 350°C with a spray voltage of 3.5 kV and sheath, auxiliary, and sweep gas flows of 45, 30, and 2, respectively. Eluted lipids were electrospray-ionized in both negative and positive modes in separate runs and fragmented by alternating higher-energy collision dissociation (HCD) and collision-induced dissociation (CID) using a precursor scan of m/z 200–2,000 at a mass resolution of 120k. This was followed by data-independent MS/MS of the top 4 ions. An isolation width of 2 m/z units and a maximum charge state of 2 were used for both CID and HCD scans. Normalized collision energies for CID and HCD were 35 and 30, respectively. CID spectra were acquired in the ion trap using an activation $Q$ value of 0.18 while HCD spectra were acquired in the Orbitrap at a mass resolution of 15k and a first fixed mass of m/z 90.

## Lipid identification and data preprocessing

The open-source in-house developed software Lipid Informed Quantitation and Identification (LIQUID) was used to identify all lipid species. This was done by importing LC-MS/MS raw data files and examining the diagnostic ion fragments and associated chain fragments in tandem mass spectra. Mass error of measured precursor, isotopic profile, and extracted ion chromatogram were also examined for determining lipid identities. A target database comprised of known lipid names, retention times, and observed m/z ratios was used to identify features across all LC-MS/MS runs by aligning

and gap-filling the mass spectrometry data of individual sample runs. MZmine 2 (68) was used to align and gap-fill all data sets by matching unidentified features to their corresponding identified features. Peak intensities were exported for statistical analysis after all aligned features were manually verified.

## Statistical analysis

Positive and negative mode lipidomic data were normalized separately. Lipid peak intensities were log2-transformed then normalized by median scaling (69). One-way analysis of variance was used to identify the most discriminant lipids across all samples. Global variations in the lipidomes of *H. congolense* WG10 samples were explored and visualized using unsupervised hierarchical clustering, employing the Statistical Analysis (one factor) module of MetaboAnalyst v5.0 (31). Lipid profiles of sample pairs were compared by statistical significance testing using Student's *t*-test and effect size evaluation using log2 fold change. Lipids were deemed significantly differential if |FC| > 1.5 and *P* < 0.05. Volcano plots and box plots were generated in RStudio running on R v4.1.2 to visualize outcomes. For each pairwise comparison, we calculated the double bond index and mean acyl chain length of the differential lipids based on fatty acid composition. These are proxies for membrane unsaturation (fluidity) and thickness (70). Essentially, log2-transformed abundance of each differential lipid was multiplied by its number of double bonds and hydrocarbon chain length. The total of these weighted abundances was then divided by the cumulative sum of lipid abundances.

The Biomarker Analysis module of MetaboAnalyst v5.0 was used to predictively model biofilm growth in *H. congolense* WG10. Instead of the classical univariate approach, we carried out multivariate exploratory receiver operating characteristic analysis, which considers relationships between the features that account for observed variances. ROC curves were generated by Monte–Carlo cross-validation (MCCV) using balanced subsampling. The most important features (lipids), as determined by ranking during each MCCV using two-thirds of the samples, were used to build classification models. The remaining one-third of the samples were used for model validation. The machine learning algorithm, random forest, was used as both the classification and feature ranking method. Furthermore, a PLS-DA model was built using the leave-one-out cross-validation method.

## ACKNOWLEDGMENTS

This work was funded by the US Department of Energy, Office of Science, Office of Biological and Environmental Research (BER), and Established Program to Stimulate Competitive Research (EPSCoR) (award number DESC0019444).

We also acknowledge Northeast Natural Energy and West Virginia University for access to the MSEEL-2 wells and use of their facilities throughout this research project. Finally, we thank the reviewers who provided very insightful comments that improved the quality and readability of the manuscript.

Conceptualization was performed by P.J.M.; data curation was performed by K.B. and C.J.U.; formal analysis was performed by C.J.U.; funding acquisition was performed by P.J.M.; investigation was performed by F.C., J.A., K.B., S.A.W., and J.E.; methodology was performed by P.J.M., F.C., and J.A.; project administration was performed by P.J.M.; visualization was performed by C.J.U. and P.J.M.; writing was performed by C.J.U.; and writing (review and editing) was performed by C.J.U., F.C., J.A., K.B., S.A.W., J.E., and P.J.M.

## AUTHOR AFFILIATIONS

[1]Natural Resources and Earth Systems Science, University of New Hampshire, Durham, New Hampshire, USA

[2]Department of Civil and Environmental Engineering, University of New Hampshire, Durham, New Hampshire, USA

[3]New England Biolabs, Ipswich, Massachusetts, USA

[4]Tetra Tech Inc., King of Prussia, Pennsylvania, USA

[5]Biological Sciences Division, Pacific Northwest National Laboratory, Richland, Washington, USA

## AUTHOR ORCIDs

Chika Jude Ugwuodo http://orcid.org/0000-0001-7184-8015

Paula J. Mouser http://orcid.org/0000-0003-2316-0915

## FUNDING

| Funder | Grant(s) | Author(s) |
|---|---|---|
| U.S. Department of Energy (DOE) | DESC0019444 | Chika Jude Ugwuodo |
| | | Fabrizio Colosimo |
| | | Jishnu Adhikari |
| | | Kent Bloodsworth |
| | | Stephanie A. Wright |
| | | Josie Eder |
| | | Paula J. Mouser |

## AUTHOR CONTRIBUTIONS

Chika Jude Ugwuodo, Data curation, Formal analysis, Software, Validation, Visualization, Writing – original draft, Writing – review and editing | Fabrizio Colosimo, Investigation, Methodology, Project administration, Validation, Writing – review and editing | Jishnu Adhikari, Investigation, Methodology, Project administration, Validation, Writing – review and editing | Kent Bloodsworth, Data curation, Investigation, Software, Writing – review and editing | Stephanie A. Wright, Investigation, Writing – review and editing | Josie Eder, Investigation, Writing – review and editing | Paula J. Mouser, Conceptualization, Funding acquisition, Methodology, Project administration, Resources, Supervision, Validation, Visualization, Writing – review and editing

## DATA AVAILABILITY

All mass spectrometry lipid data were deposited in NEXUS (Project Number: 51545), the public user portal for Environmental Molecular Sciences Laboratory (EMSL), Pacific Northwest National Laboratory (PNNL), and are available at https://dx.doi.org/10.25582/data.2023-11.2949061/2216895.

## ADDITIONAL FILES

The following material is available online.

### Supplemental Material

**Figure S1 (Spectrum02334-23-s0001.tiff).** Phosphatidylcholine turnover.
**Figure S2 (Spectrum02334-23-s0002.tiff).** Heatmap of discriminant lipids.
**Figure S3 (Spectrum02334-23-s0003.tiff).** Validation of models.
**Figure S4 (Spectrum02334-23-s0004.tiff).** Double bond and chain length variations.
**Supplementary Appendix Text (Spectrum02334-23-s0005.docx).** Supplementary text and tables.

### Open Peer Review

**PEER REVIEW HISTORY (review-history.pdf).** An accounting of the reviewer comments and feedback.

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
