## [Reviewer comments · Microbiology Spectrum]

Microbiology Spectrum

Changes in environmental and engineered conditions alter the plasma membrane lipidome of fractured shale bacteria

Chika Ugwuodo, Fabrizio Colosimo, Jishnu Adhikari, Kent Bloodsworth, Stephanie Wright, Josie Eder, and Paula J. Mouser

Corresponding Author(s): Paula J. Mouser, University of New Hampshire

Review Timeline:

Submission Date:	June 5, 2023
Editorial Decision:	August 10, 2023
Revision Received:	September 27, 2023
Editorial Decision:	October 21, 2023
Revision Received:	October 27, 2023
Accepted:	October 31, 2023

Editor: Stacey Gilk

Reviewer(s): Disclosure of reviewer identity is with reference to reviewer comments included in decision letter(s). The following individuals involved in review of your submission have agreed to reveal their identity: Genevieve Kahrilas (Reviewer #1); Christian Sohlenkamp (Reviewer #2)

Transaction Report:

DOI: <https://doi.org/10.1128/spectrum.02334-23>

August 10, 2023

Prof. Paula J. Mouser
University of New Hampshire
Civil and Environmental Engineering
35 Colovos Rd
236 Gregg Hall
Durham, NH 03824

Re: Spectrum02334-23 (Environmental and engineered changes induce plasma membrane intact polar lipid chemistry adjustments in fractured shale microbes)

Dear Prof. Paula J. Mouser:

Thank you for submitting your manuscript to Microbiology Spectrum. The reviews from two experts in the field can be found below. Both reviewers felt your study was interesting and makes an important contribution to the field; however, both expressed questions regarding analysis of the mass spec data and expanding both the introduction and discussion.

Link Not Available

Sincerely,

Stacey Gilk

Journals Department
Reviewer comments:

Reviewer #1 (Comments for the Author):

Summary and General Impressions

The research reviewed studies the changes in membrane intact polar lipid (IPL) composition in microorganisms in fractured shale reservoirs in response to high salinity and during biofilm formation. The authors find that increased salinity in the extracellular environment results in an increase in anionic IPLs, and biofilm growth in salinity corresponds to an increase in zwitterionic IPLs (where anionic IPLs remain elevated). These findings suggest a means by which performing a lipidome profile

on bacteria found within produced water may be used to detect biofilm formation downhole (by watching for large increases in zwitterionic PE). Biofilm growth downhole is otherwise very hard to detect, making this research potentially very valuable to the fracking industry.

In general, I enjoyed the paper and believe it makes a significant contribution to the existing body of research. My major concerns center around how datasets collected by negative vs. positive ionization were compared, and some overstatement in terms of biocidal efficacy resulting from altered membrane IPL composition.

Specific comments, major

1. Materials and Methods, Field Sampling / figure 8: Reading the methods, it seems that the flowback/produced water collected was from several different fractured wells. Were the conditions between all these wells homogenous enough to draw conclusions about external events affecting the membrane IPL changes? Specific questions:

- a. Were all of these wells fracked at the same time?
- b. Was the pressure released on all of the wells at the same time? (in other words, was the first day flowback was collected the same for all wells?)
- c. Were the downhole conditions (temp, salinity, pressure) similar between the wells?
- d. Was the same hydraulic fracturing fluid chemistry used at all the wells?
- e. Was the water used to frack the wells sourced from the same location?
- f. Was the proppant/sand used at all the wells the same? (proppant is a major mode of bacterial introduction into fracked wells)

2. Materials and Methods, Growth Experiments: At what pH were the cultures grown and experiments carried out? Rationale: changes in extracellular pH are also known to modulate membrane IPL composition. Fracking fluid is all pH adjusted (acidic), and flowback water/produced water then rises in pH as the fracking fluid is washed out and the formation's intrinsic minerals/carbonates control the water's pH. Additionally, the extreme ionic strength of produced water wreaks havoc on the water's pH; highly concentrated solutions result in activity coefficients increasing past ionic strengths of about 0.5M, sometimes resulting in activity coefficients greater than 1 in highly saline solutions, meaning that high NaCl concentrations actually result in pH changes even without addition of acid/base. Therefore, I think reporting the pH of the experiments would be a highly useful data point.

3. Results, paragraph 2 / materials and methods, Mass Spectrometry Analysis / Statistical Analysis: My concern centers around how the data from positive ion mode (PIM) was combined with the data from negative ion mode (NIM). Both positive and negative ion modes were used to measure different specific lipids, and then each was "normalized separately," putting each set of data on its own relative scale. How, then, were those two data sets compared and combined? Without some sort of absolute concentration to provide a point of reference between the data from PIM and NIM, or assuming the total number of lipids analyzed by PIM and NIM are always equal (cannot be true since your results show that the lipids seen by NIM - anionic lipids and PE - increase in high salinity, while those detected by PIM - neutrals and some zwitterionic lipids - decrease), it is unclear to me how data collected by PIM was compared to data collected by NIM.

4. Discussion, last paragraph: two points regarding the discussion on biocides

- a. Mentions that glutaraldehyde (GA) and quaternary ammonium compounds (QACs) both directly target the plasma membrane or use it as a conduit to access the intracellular space. The latter, to my knowledge, is not true. QACs are correctly described, but GA is a highly reactive molecule that nonspecifically reacts with proteins, and its mode of action is overwhelmingly acting on proteins in the cell wall surface. There is little chance that GA could enter into a bacterium without reacting with something else first.
- b. Careful not to overstate findings with regards to biocide efficacy. Though efficacy might certainly change, I would caution against stating that the efficacy would be limited without proper citation or follow-up experiments in the correct hypersaline, temperature-controlled, pH-adjusted environment. For example, GA's reactive intermediate is actually stabilized by presence of salt, potentially making it slightly less reactive and less effective in salinity. QACs and other lytic biocides, while they do depend on electrostatic interactions to become attracted to the cell wall surface, exert their biocidal action only when the carbon chain tail intercalates into the plasma membrane, a process mediated by hydrophobicity, which is potentially enhanced in saline environments. Therefore, while extremely interesting and deserving of further study, the overall effect on biocidal efficacy cannot be stated without further experimentation.

Specific comments, minor

5. Introduction, paragraph 3: It might enhance clarity if neutral/anionic/zwitterionic classification of plasma membrane lipids was stated as lipids are introduced

6. Discussion, last paragraph: Acronym definition (QAC) left out after mention of full name (quaternary ammonium compounds).

7. Discussion: May be relevant to discuss that temperature also results in membrane IPL changes. (Temp up, polar lipids up.) Various shale formations have very different downhole temperatures; though the Marcellus is quite mild, formations in Texas (Eagle Ford for example) are extremely deep and extremely hot.

8. Discussion: May also be useful to mention that pressure may play a part as well. Fracking conditions downhole expose the microorganisms to extremely high pressure which is also known to modulate bacterial membrane lipid composition.

9. Figure 1, pane B: May be helpful to show the correct charges in the chemical structures; only PC has its correct zwitterionic form shown.

Point of Interest (no change requested)

10. With regards to your contradictory finding about PE increasing due to salinity when prior studies note that it decreases: I find it interesting that this zwitterionic lipid was detected using negative ion mode (NIM), as opposed to the other zwitterionic lipid PC which was detected using positive ion mode (PIM). This means that PE has far more negative charges than PC (despite having a net charge of 0) since it showed a clear NIM signal. Included this comment just in case it was useful to your thinking.

Reviewer #2 (Public repository details (Required)):

I assume that the MS data could or should be submitted to a database. Not sure what the policy of Spectrum is.

Reviewer #2 (Comments for the Author):

The present manuscript by Ugwuodo et al describes the characterization of the intact polar lipid composition of the bacterium *Halanaerobium congolense* WG10 and mixed microbial consortia enriched from shale-produced fluids under different experimental conditions. Adjustments in lipid composition were observed when comparing planktonic-grown vs biofilm-grown cells and when comparing cells grown under different salinity conditions.

The manuscript presents interesting data, but personally, I find that the introductory part is much too short. By reading only the present manuscript, it is not clear what the problem of bacterial growth during the fracking process is. In this respect, I recommend enriching the introduction, like what is written in a 2022 paper of the same group published in *Frontiers in Microbiology*.

General comments:

- 1) I noticed that the genome of *H. congolense* is sequenced and personally I would very much appreciate it if the authors could discuss what genes involved in IPL synthesis, turnover, and other physiological processes mentioned throughout are present in the genome and possibly responsible for the observations.
- 2) It is not clear to me how the hydraulic retention time experiments are made.
- 3) It is interesting that the authors analyze the quantities of each single lipid subspecies separately, but looking at the results and I am not sure that many conclusions can be made from the data if analyzed exclusively this way. The authors should also try other types of analysis for example to cluster together lipid species with similar fatty acid compositions. I am not convinced that drastic changes in a single specific subspecies are enough to change the membrane properties as required for bacterial adaptation to stress conditions. A large part of the results section is a description of the behavior of single lipid subspecies, and I think the authors are getting lost in the details.

Comments:

- 1) The paragraph "Importance" should be rewritten to make it easier to understand the problems that can be caused by the presence of bacteria during the fracking process.
- 2) In the introduction the authors write that the genus "*Halanerobium* dominates the persisting microbial communities in geologically distinct shale reservoirs...". When comparing the results of the IPL composition analysis, this does not seem the case. IPL composition of *Halanerobium* is very different from the mixed microbial consortia, thereby contradicting (in my opinion) the earlier sentence.
- 3) In the third paragraph of the introduction, the authors mention the most common bacterial phospholipids, but I don't agree with the claim that PS and PC are common. A few years ago, it was estimated that about 10% of bacteria could form PC under certain conditions. On the other hand, it was estimated that about 50% of bacteria could form ornithine lipids and other amino lipids. Please check this and modify the text.
- 4) Looking at the fourth paragraph of the introduction I think the authors are generalizing too much. They write that "evidence suggests that bacteria increase the amount of negatively charged intact polar lipids in their plasma membrane in response to increased salinities". I remember a few publications studying this in *Escherichia coli*, but we now know that there is a lot of lipid diversity in microorganisms, and we also know that *E. coli* is not always representative of other or all other bacteria. This applies also to other sections of the text, where similar generalizations are made from a few studies in one organism.
- 5) DG concentrations are very high, like 40%. This is highly unusual. Are there data from related bacteria or other data from your group with similar results? Can you discuss this or are there references for bacteria with similar high DG content?
- 6) The authors use several representations and analyses usually used in genome/transcriptome analysis which I find very interesting (volcano plots, clustering, etc.), but I am not convinced that the accumulation of a single lipid subspecies and not of other subspecies of the same lipid class is necessarily super important, for example, compare induction of gene expression to

the increased accumulation of a type of DG.

7) Fig. 1: I would recommend adding the name (abbreviations) to the structures. It is easier to see which structure is which.

8) I would recommend changing the title of the manuscript. I think it is very technical and it is not clear from the title, what the manuscript is about.

9) In the discussion the authors write about a possible role for PC as a precursor for choline and glycine betaine. They also speculate that choline is accumulating in the bacteria under saline stress. Did the authors try to detect these osmolytes?

10) I think the authors should discuss in more detail the results of their 2022 paper in the context of this new publication. The experiments and the obtention of the samples are similar so this could be a very enriching discussion.

Staff Comments:

Preparing Revision Guidelines

Please return the manuscript within 60 days; if you cannot complete the modification within this time period, please contact me. If you do not wish to modify the manuscript and prefer to submit it to another journal, please notify me of your decision immediately so that the manuscript may be formally withdrawn from consideration by Microbiology Spectrum.

Reviewer response to: “Environmental and engineered changes induce plasma membrane intact polar lipid chemistry adjustments in fractured shale microbes” by Ugwuodo

PI: Paula Mouser

Submission Journal: Microbiology Spectrum

Summary and General Impressions

The research reviewed studies the changes in membrane intact polar lipid (IPL) composition in microorganisms in fractured shale reservoirs in response to high salinity and during biofilm formation. The authors find that increased salinity in the extracellular environment results in an increase in anionic IPLs, and biofilm growth in salinity corresponds to an increase in zwitterionic IPLs (where anionic IPLs remain elevated). These findings suggest a means by which performing a lipidome profile on bacteria found within produced water may be used to detect biofilm formation downhole (by watching for large increases in zwitterionic PE). Biofilm growth downhole is otherwise very hard to detect, making this research potentially very valuable to the fracking industry.

In general, I enjoyed the paper and believe it makes a significant contribution to the existing body of research. My major concerns center around how datasets collected by negative vs. positive ionization were compared, and some overstatement in terms of biocidal efficacy resulting from altered membrane IPL composition.

Specific comments, major

1. Materials and Methods, Field Sampling / figure 8: Reading the methods, it seems that the flowback/produced water collected was from several different fractured wells. Were the conditions between all these wells homogenous enough to draw conclusions about external events affecting the membrane IPL changes? Specific questions:
 - a. Were all of these wells fracked at the same time?
 - b. Was the pressure released on all of the wells at the same time? (in other words, was the first day flowback was collected the same for all wells?)
 - c. Were the downhole conditions (temp, salinity, pressure) similar between the wells?
 - d. Was the same hydraulic fracturing fluid chemistry used at all the wells?
 - e. Was the water used to frack the wells sourced from the same location?
 - f. Was the proppant/sand used at all the wells the same? (proppant is a major mode of bacterial introduction into fracked wells)

2. Materials and Methods, Growth Experiments: At what pH were the cultures grown and experiments carried out? Rationale: changes in extracellular pH are also known to modulate membrane IPL composition. Fracking fluid is all pH adjusted (acidic), and flowback water/produced water then rises in pH as the fracking fluid is washed out and the formation’s intrinsic minerals/carbonates control the water’s pH. Additionally, the extreme ionic strength of produced water wreaks havoc on the water’s pH; highly

concentrated solutions result in activity coefficients *increasing* past ionic strengths of about 0.5M, sometimes resulting in activity coefficients greater than 1 in highly saline solutions, meaning that high NaCl concentrations actually result in pH changes even without addition of acid/base. Therefore, I think reporting the pH of the experiments would be a highly useful data point.

3. Results, paragraph 2 / materials and methods, Mass Spectrometry Analysis / Statistical Analysis: My concern centers around how the data from positive ion mode (PIM) was combined with the data from negative ion mode (NIM). Both positive and negative ion modes were used to measure different specific lipids, and then each was “normalized separately,” putting each set of data on its own relative scale. How, then, were those two data sets compared and combined? Without some sort of absolute concentration to provide a point of reference between the data from PIM and NIM, or assuming the total number of lipids analyzed by PIM and NIM are always equal (cannot be true since your results show that the lipids seen by NIM – anionic lipids and PE – increase in high salinity, while those detected by PIM – neutrals and some zwitterionic lipids – decrease), it is unclear to me how data collected by PIM was compared to data collected by NIM.
4. Discussion, last paragraph: two points regarding the discussion on biocides
 - a. Mentions that glutaraldehyde (GA) and quaternary ammonium compounds (QACs) both directly target the plasma membrane or *use it as a conduit to access the intracellular space*. The latter, to my knowledge, is not true. QACs are correctly described, but GA is a highly reactive molecule that nonspecifically reacts with proteins, and its mode of action is overwhelmingly acting on proteins in the cell wall surface. There is little chance that GA could enter into a bacterium without reacting with something else first.
 - b. Careful not to overstate findings with regards to biocide efficacy. Though efficacy might certainly change, I would caution against stating that the efficacy would be *limited* without proper citation or follow-up experiments in the correct hypersaline, temperature-controlled, pH-adjusted environment. For example, GA's reactive intermediate is actually stabilized by presence of salt, potentially making it slightly less reactive and less effective in salinity. QACs and other lytic biocides, while they *do* depend on electrostatic interactions to become attracted to the cell wall surface, exert their biocidal action only when the carbon chain tail intercalates into the plasma membrane, a process mediated by hydrophobicity, which is potentially *enhanced* in saline environments. Therefore, while extremely interesting and deserving of further study, the overall effect on biocidal efficacy cannot be stated without further experimentation.

Specific comments, minor

5. Introduction, paragraph 3: It might enhance clarity if neutral/anionic/zwitterionic classification of plasma membrane lipids was stated as lipids are introduced

6. Discussion, last paragraph: Acronym definition (QAC) left out after mention of full name (quaternary ammonium compounds).
7. Discussion: May be relevant to discuss that temperature also results in membrane IPL changes. (Temp up, polar lipids up.) Various shale formations have very different downhole temperatures; though the Marcellus is quite mild, formations in Texas (Eagle Ford for example) are extremely deep and extremely hot.
8. Discussion: May also be useful to mention that pressure may play a part as well. Fracking conditions downhole expose the microorganisms to extremely high pressure which is also known to modulate bacterial membrane lipid composition.
9. Figure 1, pane B: May be helpful to show the correct charges in the chemical structures; only PC has its correct zwitterionic form shown.

Point of Interest (no change requested)

10. With regards to your contradictory finding about PE increasing due to salinity when prior studies note that it decreases: I find it interesting that this zwitterionic lipid was detected using negative ion mode (NIM), as opposed to the other zwitterionic lipid PC which was detected using positive ion mode (PIM). This means that PE has far more negative charges than PC (despite having a net charge of 0) since it showed a clear NIM signal. Included this comment just in case it was useful to your thinking.

Reviewer #1 (Comments for the Author):

Summary and General Impressions

The research reviewed studies the changes in membrane intact polar lipid (IPL) composition in microorganisms in fractured shale reservoirs in response to high salinity and during biofilm formation. The authors find that increased salinity in the extracellular environment results in an increase in anionic IPLs, and biofilm growth in salinity corresponds to an increase in zwitterionic IPLs (where anionic IPLs remain elevated). These findings suggest a means by which performing a lipidome profile on bacteria found within produced water may be used to detect biofilm formation downhole (by watching for large increases in zwitterionic PE). Biofilm growth downhole is otherwise very hard to detect, making this research potentially very valuable to the fracking industry.

In general, I enjoyed the paper and believe it makes a significant contribution to the existing body of research. My major concerns center around how datasets collected by negative vs. positive ionization were compared, and some overstatement in terms of biocidal efficacy resulting from altered membrane IPL composition. – Thank you very much. We appreciate that you found our work potentially valuable. We are very happy to address issues raised and revise accordingly.

Specific comments, major

1. Materials and Methods, Field Sampling / figure 8: Reading the methods, it seems that the flowback/produced water collected was from several different fractured wells. Were the conditions between all these wells homogenous enough to draw conclusions about external events affecting the membrane IPL changes? Specific questions:

- a. Were all of these wells fracked at the same time?
- b. Was the pressure released on all of the wells at the same time? (in other words, was the first day flowback was collected the same for all wells?)
- c. Were the downhole conditions (temp, salinity, pressure) similar between the wells?
- d. Was the same hydraulic fracturing fluid chemistry used at all the wells?
- e. Was the water used to frack the wells sourced from the same location?

f. Was the proppant/sand used at all the wells the same? (proppant is a major mode of bacterial introduction into fracked wells) – Thank you for bringing this confusion to our attention. We clarified the language in the methods section to state that several wells were sampled during the field trips but this current study used produced fluids from just one of the wells. As a result, there are no differences in geochemistry or downhole pressure that would have influenced the enrichment microbial communities. The language has been clarified in the manuscript [Materials and Methods, line 463; and Figure 8, line 842].

2. Materials and Methods, Growth Experiments: At what pH were the cultures grown and experiments carried out? Rationale: changes in extracellular pH are also known to modulate membrane IPL composition. Fracking fluid is all pH adjusted (acidic), and flowback water/produced water then rises in pH as the fracking fluid is washed out and the formation's intrinsic minerals/carbonates control the water's pH. Additionally, the extreme ionic strength of produced water wreaks havoc on the water's pH; highly concentrated solutions result in activity coefficients increasing past ionic strengths of about 0.5M, sometimes resulting in activity coefficients greater than 1 in highly saline solutions, meaning that high NaCl concentrations actually result in pH changes even without addition of acid/base. Therefore, I think reporting the pH of the experiments would be a highly useful data point. – pH was maintained at 7.0 throughout the growth experiments in our Sartorius bioreactors by automatic addition of 1 M KOH and 1 M HCl. We have included this information in the Materials and Methods section (Lines 487-488).

3. Results, paragraph 2 / materials and methods, Mass Spectrometry Analysis / Statistical Analysis: My concern centers around how the data from positive ion mode (PIM) was combined with the data from negative ion mode (NIM). Both positive and negative ion modes were used to measure different specific lipids, and then each was "normalized separately," putting each set of data on its own relative scale. How, then, were those two data sets compared and combined? Without some sort of absolute concentration to provide a point of reference between the data from PIM and NIM, or assuming the total number of lipids analyzed by PIM and NIM are always equal (cannot be true since your results show that the lipids seen by NIM - anionic lipids and PE - increase in high salinity, while those detected by PIM - neutrals and some zwitterionic lipids - decrease), it is unclear to me how data collected by PIM was compared to data collected by NIM. – Thanks so much for pointing this out, we absolutely agree with this comment. For clarity, we normalized the PIM and NIM data separately as per Kyle et al. (2019) (<https://www.pnas.org/doi/full/10.1073/pnas.1815356116>) and other prior studies. They were then combined to conduct some of the analyses. But just like you pointed out, it is questionable to compare data acquired in the PIM with those acquired in the NIM within a sample. This is why we only compared normalized abundances of species between independent samples. So, essentially, we made statements like "*compared to Sample A, PE is more abundant in Sample B.*" Statements like this were similar for the other lipids. We avoided statements like "*PE (acquired in the NIM) was more abundant than PC (acquired in the PIM) in sample A*". The latter statement would be faulty since data for both lipids were acquired differently, scaled differently, and we didn't obtain absolute concentrations.

4. Discussion, last paragraph: two points regarding the discussion on biocides

a. Mentions that glutaraldehyde (GA) and quaternary ammonium compounds (QACs) both directly target the plasma membrane or use it as a conduit to access the intracellular space. The latter, to my knowledge, is not true. QACs are correctly described, but GA is a highly reactive molecule that nonspecifically reacts with proteins, and its mode of action is overwhelmingly acting on proteins in the cell wall surface. There is little chance that GA could enter into a bacterium without reacting with something else first. – We have looked into this and revised the referenced statement to indicate that GA primarily targets cell surface amines (Discussion, lines 425-429).

b. Careful not to overstate findings with regards to biocide efficacy. Though efficacy might certainly change, I would caution against stating that the efficacy would be limited without proper citation or follow-up experiments in the correct hypersaline, temperature-controlled, pH-adjusted environment. For example, GA's reactive intermediate is actually stabilized by presence of salt, potentially making it slightly less reactive and less effective in salinity. QACs and other lytic biocides, while they do depend on electrostatic interactions to become attracted to the cell wall surface, exert their biocidal action only when the carbon chain tail intercalates into the plasma membrane, a process mediated by hydrophobicity, which is potentially enhanced in saline environments. Therefore, while extremely interesting and deserving of further study, the overall effect on biocidal efficacy cannot be stated without further experimentation. – This is very well noted and we have revised this part of the Discussion section (Lines 434-442) to avoid overstating our findings. We have highlighted that biocidal efficacy is dependent on several interacting factors and that further studies under controlled conditions are needed to make any emphatic deductions.

Specific comments, minor

5. Introduction, paragraph 3: It might enhance clarity if neutral/anionic/zwitterionic classification of plasma membrane lipids was stated as lipids are introduced – We have moved up the “lipid classification by charge” part to where microbial lipids are introduced (Lines 105-107).

6. Discussion, last paragraph: Acronym definition (QAC) left out after mention of full name (quaternary ammonium compounds). – Acronym has been added after mentioning the full name in the affected paragraph of Discussion (Line 423-424).

7. Discussion: May be relevant to discuss that temperature also results in membrane IPL changes. (Temp up, polar lipids up.) Various shale formations have very different downhole temperatures; though the Marcellus is quite mild, formations in Texas (Eagle Ford for example) are extremely deep and extremely hot. – This is very important to point out, and

we have done so in the discussion section of the manuscript (Lines 401-420). We have discussed how temperature and pressure are also important perturbants of microbial plasma membrane features.

8. Discussion: May also be useful to mention that pressure may play a part as well. Fracking conditions downhole expose the microorganisms to extremely high pressure which is also known to modulate bacterial membrane lipid composition. – We agree, and have revised the manuscript accordingly. Please see point 7 above (Discussion, Lines 401-420).

9. Figure 1, pane B: May be helpful to show the correct charges in the chemical structures; only PC has its correct zwitterionic form shown. – We have modified the structures in pane B of Figure 1 to show their correct charges.

Point of Interest (no change requested)

10. With regards to your contradictory finding about PE increasing due to salinity when prior studies note that it decreases: I find it interesting that this zwitterionic lipid was detected using negative ion mode (NIM), as opposed to the other zwitterionic lipid PC which was detected using positive ion mode (PIM). This means that PE has far more negative charges than PC (despite having a net charge of 0) since it showed a clear NIM signal. Included this comment just in case it was useful to your thinking. – Very interesting observation. Absolutely helpful and very well noted.

Reviewer #2 (Public repository details (Required)):

I assume that the MS data could or should be submitted to a database. Not sure what the policy of Spectrum is. – We have a 'Data Availability' statement in the manuscript (Lines 592-594). The MS data generated from this study can be accessed from NEXUS - Project Number: 51545.

Reviewer #2 (Comments for the Author):

The present manuscript by Ugwuodo et al describes the characterization of the intact polar lipid composition of the bacterium Halanaerobium congolense WG10 and mixed microbial consortia enriched from shale-produced fluids under different experimental conditions. Adjustments in lipid composition were observed when comparing planktonic-grown vs

biofilm-grown cells and when comparing cells grown under different salinity conditions. The manuscript presents interesting data, but personally, I find that the introductory part is much too short. By reading only the present manuscript, it is not clear what the problem of bacterial growth during the fracking process is. In this respect, I recommend enriching the introduction, like what is written in a 2022 paper of the same group published in *Frontiers in Microbiology*. – We have further expanded the introduction, expanding on the problems bacteria cause in fractured shale and improving our establishment of the study's motivation (Introduction, lines 62-82).

General comments:

1) I noticed that the genome of *H. congolense* is sequenced and personally I would very much appreciate it if the authors could discuss what genes involved in IPL synthesis, turnover, and other physiological processes mentioned throughout are present in the genome and possibly responsible for the observations. – Thank you for this comment. We have now incorporated this idea throughout the discussion section. For example, see highlighted the genes in the published *H. congolense* genome potentially responsible for anionic phospholipid synthesis (Lines 296-299), phosphocholine turnover (Lines 323-326) and glycine betaine reduction (Lines 342-345).

2) It is not clear to me how the hydraulic retention time experiments are made. – We have clarified the hydraulic retention time experiments in the Materials and Methods section under "Planktonic growth experiments" (Lines 477-482).

3) It is interesting that the authors analyze the quantities of each single lipid subspecies separately, but looking at the results and I am not sure that many conclusions can be made from the data if analyzed exclusively this way. The authors should also try other types of analysis for example to cluster together lipid species with similar fatty acid compositions. I am not convinced that drastic changes in a single specific subspecies are enough to change the membrane properties as required for bacterial adaptation to stress conditions. A large part of the results section is a description of the behavior of single lipid subspecies, and I think the authors are getting lost in the details. – We absolutely agree that it is important to capture the bigger picture as well. This is why our analysis involved first, identifying the differentially expressed lipid species, which is very important. Then grouping them by subclass and highlighting their charge (anionic, zwitterionic and neutral) variations. Additionally, based on your recommendation, we have gone ahead to analyze the bulk fatty acid features (mean chain length and double bond index) of the discriminant lipids for all comparisons (Statistical analysis, lines 561-566). The outcomes of this analysis have been integrated into the results narratives (for examples, lines 202-206, 218-221, 239-243, etc.).

Also see the new supplementary Figure S4. Results of this new analysis are discussed in the supplementary appendix (Lines 871-892).

Comments:

1) The paragraph "Importance" should be rewritten to make it easier to understand the problems that can be caused by the presence of bacteria during the fracking process. – Thanks for this comment, we have revised the importance statement accordingly to communicate the problems bacteria cause in fractured wells more clearly to the reader.

2) In the introduction the authors write that the genus "Halanaerobium dominates the persisting microbial communities in geologically distinct shale reservoirs...". When comparing the results of the IPL composition analysis, this does not seem the case. IPL composition of Halanaerobium is very different from the mixed microbial consortia, thereby contradicting (in my opinion) the earlier sentence. – We have revised this statement in the introduction section (Lines 62-67) to note that although *Halanaerobium* dominates the microbiome of several wells, there are exceptions as community composition depends on several factors including intrinsic reservoir geochemistry, well operation and the timepoint of analysis.

3) In the third paragraph of the introduction, the authors mention the most common bacterial phospholipids, but I don't agree with the claim that PS and PC are common. A few years ago, it was estimated that about 10% of bacteria could form PC under certain conditions. On the other hand, it was estimated that about 50% of bacteria could form ornithine lipids and other amino lipids. Please check this and modify the text. – Thank you for pointing out this contradiction. We have revised this part of the introduction section accordingly (Lines 108-115).

4) Looking at the fourth paragraph of the introduction I think the authors are generalizing too much. They write that "evidence suggests that bacteria increase the amount of negatively charged intact polar lipids in their plasma membrane in response to increased salinities". I remember a few publications studying this in *Escherichia coli*, but we now know that there is a lot of lipid diversity in microorganisms, and we also know that *E. coli* is not always representative of other or all other bacteria. This applies also to other sections of the text, where similar generalizations are made from a few studies in one organism. – We have revised these overgeneralizing statements across all the affected sections of the text. This includes (1) generalization about *Halanaerobium* dominance (Introduction, lines 62-67), (2) statement about common IPL lipids in bacteria (Introduction, lines 108-115), (3) generalization of intact polar lipid chemistry response to high salinities (Introduction, lines

116-121), and (4) statement on phospholipase expression in “most” bacteria (Discussion, line 321).

5) DG concentrations are very high, like 40%. This is highly unusual. Are there data from related bacteria or other data from your group with similar results? Can you discuss this or are there references for bacteria with similar high DG content? – We agree that this is definitely unusual and we have discussed why this might be the case and highlighted the need for further investigation into this (see Results, lines 173-175; and Supplementary Appendix, lines 854-870).

6) The authors use several representations and analyses usually used in genome/transcriptome analysis which I find very interesting (volcano plots, clustering, etc.), but I am not convinced that the accumulation of a single lipid subspecies and not of other subspecies of the same lipid class is necessarily super important, for example, compare induction of gene expression to the increased accumulation of a type of DG. – Please see Point 3 above under General Comments where we have attempted to address this issue.

7) Fig. 1: I would recommend adding the name (abbreviations) to the structures. It is easier to see which structure is which. – We have added abbreviations to the structures in Figure 1.

8) I would recommend changing the title of the manuscript. I think it is very technical and it is not clear from the title, what the manuscript is about. – We have revised the title of the manuscript to make it more descriptive of our study and easier to understand

9) In the discussion the authors write about a possible role for PC as a precursor for choline and glycine betaine. They also speculate that choline is accumulating in the bacteria under saline stress. Did the authors try to detect these osmolytes? – For this study, we did not try to detect these osmolytes. We have more cautiously revised these speculative statements about PC recycling in the Discussion section (Lines 323-328). We have highlighted the need for proteomic and metabolomic evidence to substantiate these speculations.

10) I think the authors should discuss in more detail the results of their 2022 paper in the context of this new publication. The experiments and the obtention of the samples are similar so this could be a very enriching discussion. – Thank you for suggesting this. We have done this in a new paragraph in the supplementary appendix (Lines 871-892).

October 21, 2023

Prof. Paula J. Mouser
University of New Hampshire
Civil and Environmental Engineering
35 Colovos Rd
236 Gregg Hall
Durham, NH 03824

Re: Spectrum02334-23R1 (Changes in environmental and engineered conditions alter the plasma membrane lipidome of fractured shale bacteria)

Dear Prof. Paula J. Mouser:

Thank you for submitting your manuscript to Microbiology Spectrum. As you will see your paper is very close to acceptance. Please modify the manuscript along the lines as recommended by Reviewer 2, primarily checking the PC structure in Supplemental Figure 1.

As these revisions are quite minor, I expect that you should be able to turn in the revised paper in less than 30 days, if not sooner. If your manuscript was reviewed, you will find the reviewers' comments below.

When submitting the revised version of your paper, please provide (1) point-by-point responses to the issues raised by the reviewers as file type "Response to Reviewers," not in your cover letter, and (2) a PDF file that indicates the changes from the original submission (by highlighting or underlining the changes) as file type "Marked Up Manuscript - For Review Only". Please use this link to submit your revised manuscript. Detailed instructions on submitting your revised paper are below.

Link Not Available

Sincerely,

Stacey Gilk

Reviewer comments:

Reviewer #2 (Comments for the Author):

The present manuscript by Ugwuodo et al describes the characterization of the intact polar lipid composition of the bacterium *Halanaerobium congolense* WG10 and mixed microbial consortia enriched from shale-produced fluids under different experimental conditions. Adjustments in lipid composition were observed when comparing planktonic-grown vs biofilm-grown cells and when comparing cells grown under different salinity conditions. This is a revised version of a manuscript submitted a few months earlier. In my opinion the manuscript improved a lot and is far more accessible to readers with a general interest in microbiology.

Two more comments:

- 1) There are still a few typos in the text, please check.
- 2) The PC structure on top of Figure S1 is not correct. Check the phosphate group.

Preparing Revision Guidelines

Please return the manuscript within 60 days; if you cannot complete the modification within this time period, please contact me. If you do not wish to modify the manuscript and prefer to submit it to another journal, please notify me of your decision immediately so that the manuscript may be formally withdrawn from consideration by Microbiology Spectrum.

Reviewer comments:

Reviewer #2 (Comments for the Author):

The present manuscript by Ugwuodo et al describes the characterization of the intact polar lipid composition of the bacterium *Halanaerobium congolense* WG10 and mixed microbial consortia enriched from shale-produced fluids under different experimental conditions. Adjustments in lipid composition were observed when comparing planktonic-grown vs biofilm-grown cells and when comparing cells grown under different salinity conditions. This is a revised version of a manuscript submitted a few months earlier. In my opinion the manuscript improved a lot and is far more accessible to readers with a general interest in microbiology.

Two more comments:

- 1) There are still a few typos in the text, please check. – We have checked and fixed all identified typos (see Lines 126, 233, 237, 395, 518, 573, 851, 890)
- 2) The PC structure on top of Figure S1 is not correct. Check the phosphate group. – A missing oxygen atom has been added to Figure S1.

Re: Spectrum02334-23R2 (Changes in environmental and engineered conditions alter the plasma membrane lipidome of fractured shale bacteria)

Dear Prof. Paula J. Mouser:

Your manuscript has been accepted, and I am forwarding it to the ASM production staff for publication. Your paper will first be checked to make sure all elements meet the technical requirements. ASM staff will contact you if anything needs to be revised before copyediting and production can begin. Otherwise, you will be notified when your proofs are ready to be viewed.

Sincerely,
Stacey Gilk
Editor
Microbiology Spectrum